# Cellular mechanisms of heterogeneity in *NF2*-mutant schwannoma

Christine Chiasson-MacKenzie [1,2], Jeremie Vitte [3], Ching-Hui Liu[1,2], Emily A. Wright[1,2], Elizabeth A. Flynn[1,4], Shannon L. Stott[1,4], Marco Giovannini [3] & Andrea I. McClatchey [1,2] ✉

Schwannomas are common sporadic tumors and hallmarks of familial neurofibromatosis type 2 (NF2) that develop predominantly on cranial and spinal nerves. Virtually all schwannomas result from inactivation of the *NF2* tumor suppressor gene with few, if any, cooperating mutations. Despite their genetic uniformity schwannomas exhibit remarkable clinical and therapeutic heterogeneity, which has impeded successful treatment. How heterogeneity develops in *NF2*-mutant schwannomas is unknown. We have found that loss of the membrane:cytoskeleton-associated NF2 tumor suppressor, merlin, yields unstable intrinsic polarity and enables $Nf2^{-/-}$ Schwann cells to adopt distinct programs of ErbB ligand production and polarized signaling, suggesting a self-generated model of schwannoma heterogeneity. We validated the heterogeneous distribution of biomarkers of these programs in human schwannoma and exploited the synchronous development of lesions in a mouse model to establish a quantitative pipeline for studying how schwannoma heterogeneity evolves. Our studies highlight the importance of intrinsic mechanisms of heterogeneity across human cancers.

Schwannomas account for a large proportion of sporadic nervous system tumors in humans and are hallmarks of the inherited tumor predisposition syndrome neurofibromatosis type 2 (NF2)[1–4]. Although usually benign, schwannomas develop predominantly on spinal nerve roots and cranial nerves, causing significant neurological deficit, chronic pain, and morbidity. Few targeted therapies have been developed to treat schwannomas and instead, high-risk surgical removal is often necessary. This is particularly difficult for NF2 patients, who frequently develop multiple, recurring tumors.

Whether inherited or sporadic, virtually all schwannomas are caused by genetic inactivation of the *NF2* tumor suppressor gene, and few cooperating mutations have been identified[5]. Despite being genetically 'cold', schwannomas exhibit surprising intertumoral heterogeneity, including remarkably variable natural histories, growth rates, and clinical impact, even within the same NF2 patient[1]. Neither growth rate nor tumor burden correlates with the ability of schwannomas to cause pain or nerve dysfunction, and the few rational therapies that have been evaluated for schwannoma have yielded at best a heterogeneous and cytostatic response, often followed by accelerated tumor regrowth after drug cessation[6–8]. An understanding of the molecular basis of schwannoma heterogeneity is essential for the development of successful non-surgical therapies.

Intratumoral heterogeneity is evident histologically and ultrastructurally in schwannoma[9–11]. Hematoxylin and eosin staining of schwannoma tissue reveals regions of spindle-shaped cells that form concentric whorls or aligned arrays, and others of more radially-shaped cells with shorter cell processes. Ultrastructurally, cells within schwannomas exhibit striking variation in the extent of cell-cell versus

[1]Massachusetts General Hospital Cancer Center, Harvard Medical School, 149 13th Street, Charlestown, MA 02129, USA. [2]Department of Pathology, Massachusetts General Hospital, Harvard Medical School, 55 Fruit Street, Boston, MA 02114, USA. [3]Department of Head and Neck Surgery, David Geffen School of Medicine at UCLA and Jonsson Comprehensive Cancer Center (JCCC), University of California Los Angeles, Los Angeles, CA 90095, USA. [4]Center for Engineering in Medicine and BioMEMS Resource Center, Surgical Services, Massachusetts General Hospital, Harvard Medical School, 114 16th Street, Charlestown, MA 02129, USA. ✉e-mail: mcclatch@helix.mgh.harvard.edu

cell-basal lamina contact, with opposing cell surfaces coated with dense basal lamina or forming interlocking cell-cell contacts in widely varying proportions, and some regions featuring denuded basal lamina, prominent vacuoles, and microcysts[9]. Such 'Antoni A and B' regions exhibit different patterns of immune infiltration and vascularity, suggesting that they harbor distinct paracrine environments[12]. These visible features of cellular heterogeneity vary from tumor to tumor as features of intertumoral heterogeneity, but it is not known how they are established within otherwise genetically homogeneous tumors.

Schwann cells (SCs) are uniquely polarized epithelial cells that can transition between a bipolar, migratory state during development and wound repair, and an apicobasally polarized state that relies on heterotypic contact with the neuron as an extrinsic spatial polarity cue[13–15]. Rather than the discrete junctional and fluid-exposed apical surfaces of conventional epithelia, apicobasally polarized SCs form a hybrid 'apicojunctional' adaxonal surface that is dedicated to contacting the neuron, enriched in both apical and cell-cell adhesion proteins, and a crucial site of neuroglial signaling between the two cell types[16–20]. The remaining abaxonal surface contacts basal lamina that is generated by the SCs themselves. Importantly, signaling at the two SC surfaces is distinct[14]. For example, neuregulin 1 (Nrg1) that is tethered to the neuronal plasma membrane activates ErbB3 on the adaxonal SC surface to induce mammalian target of rapamycin complex 1 (mTORC1)-triggered phosphorylation of S6rp and expansion of the neuron-ensheathing membrane edge, while laminin signaling to α6-containing integrins promotes mTORC2-dependent activation of serum and glucocorticoid-induced kinase 1 (SGK1) and phosphorylation of its effector NDRG1 abaxonally[14, 21–23]. Notably, laminin-activated feedback limits Nrg1−ErbB3-driven membrane production at the adaxonal surface, suggesting that the size of the two signaling domains is normally under feedback control via unknown mechanisms[21]. In vivo SC polarity is initiated and spatially defined by contact with the neuronal cell membrane but in vitro it is imposed and sized by attachment to a laminin-coated culture dish. In schwannomas, the spatial cues that normally govern polarized signaling are lost, and cells exhibit strikingly variable polarized surface content.

The NF2-encoded tumor suppressor, merlin, and closely related plasma membrane:cytoskeleton linking ERM proteins (ezrin, radixin, moesin) share interdependent roles in organizing the cell cortex and govern intrinsic cell polarity in many contexts[24–31]. For example, in single matrix-embedded epithelial cells, merlin deficiency impairs the spatial restriction of cortical ezrin that normally polarizes the cell; instead, ectopic cortical ezrin drives unstable intrinsic polarity[27]. The importance of controlling ERM activity is underscored by recent studies demonstrating the role of ERM-mediated membrane:cytoskeleton linkage in gating stem cell pluripotency[32, 33]. Consistent with such fundamental cell cortex-organizing activities, merlin and the ERMs influence many signaling programs, including the activity of small GTPases of the Rho and Ras families, the Hippo signaling network, and mTOR kinases[34–38]. Merlin/ERM proteins also modulate the cytoskeletal response to members of the ErbB family of tyrosine kinase receptors[39–45]. For example, in liver epithelial cells merlin deficiency enhances the excitability of the cortical cytoskeleton to Epidermal Growth Factor (EGF) stimulation in an ezrin-dependent manner, enabling macropinocytosis, an actin-based mechanism of nutrient scavenging that is initiated by large cell surface ruffles[40, 46]. Through fundamental roles in organizing the cell cortex, merlin, and the ERM proteins are poised to enable the coordination of intrinsic polarity with other fundamental cellular activities.

We have found that NF2-deficient SCs exhibit unstable polarity and can adopt distinct phenotypic states featuring coordinated auto/paracrine ErbB ligand and polarity gene expression and polarized cytoskeletal organization, suggesting a model of self-generated heterogeneity that could explain the notoriously variable clinical and therapeutic behaviors of schwannomas. Importantly, this model also suggests new biomarkers of schwannoma heterogeneity that we validated in vivo using a quantitative imaging pipeline that we have developed. Our findings provide mechanistic insight into the adaptive biology of SCs, new biomarkers, and a quantitative imaging platform that can be developed to guide the treatment of human schwannoma; they also inform studies of intrinsic heterogeneity in other tumor types.

## Results

### Nf2-deficient SCs exhibit altered intrinsic polarity and exaggerated cortical responses to Nrg1

When cultured on laminin-coated tissue culture dishes, $Nf2^{flox/flox}$ (WT) SCs are bipolar with small ruffles forming at the poles, reflecting intrinsic polarity that is independent of cues imposed by axonal contact (Fig. 1a, b). In contrast, three different primary mouse $Nf2^{flox/flox}$ SC populations in which Nf2 has been acutely deleted (hereafter denoted $Nf2^{-/-}$; Supplementary Fig. 1A) exhibited a loss of this intrinsic polarity, instead featuring multiple dynamically extending processes containing larger ruffles, as has been reported (Fig. 1a, b)[47]. Multipolarity in $Nf2^{-/-}$ SCs is accompanied by a more radial cell shape and distribution of actin, compared to the long parallel stress fibers exhibited by bipolar WT SCs (Fig. 1c; Supplementary Fig. 1B). This is consistent with the fundamental role for merlin in establishing intrinsic cortical polarity that has been observed in many cell types[27, 29].

Nrg1 regulates SC proliferation, migration, and myelination, and is normally included in SC culture medium[13,48,49]. When deprived of Nrg1, $Nf2^{-/-}$ SCs exhibited a striking enhancement of basally positioned and radially distributed ezrin-containing microvilli relative to WT SCs (Fig. 1d, e, Supplementary Fig. 1C)[50]. On the other hand, acute stimulation of $Nf2^{-/-}$ SCs with Nrg1 yielded a dramatic loss of microvilli and reorganization of ezrin and actin to form a circumferential band from which large cortical ruffles emanated (Fig. 1d–g; Supplementary Fig. 1B, C). WT SCs exhibit only weak cortical ruffles and ezrin relocalization in response to Nrg1, despite activating similar levels of downstream signals (Fig. 1d–g; Supplementary Fig. 1B, D). Depletion of ezrin eliminated both multipolarity and the exaggerated cytoskeletal responses to Nrg1 availability in $Nf2^{-/-}$ SCs (Fig. 1h; Supplementary Fig. 1E−G)[40]. These data are consistent with a model wherein in $Nf2^{-/-}$ SCs, ectopic cortical ezrin drives multipolarity and distinct exaggerated cytoskeletal responses to the presence (ruffles) or absence (microvilli) of Nrg1, enabling heterogeneous cytoskeletal states.

### Progressive junctional ruffling and adhesion in $Nf2^{-/-}$ SCs in the presence of Nrg1

Like the axon-facing apicojunctional surface of normal SCs in vivo, cortical ruffles exhibited by $Nf2^{-/-}$ SCs in the presence of Nrg1 contain the adherens junction proteins N-cadherin and β-catenin, the Nrg1 receptor ErbB3 and its heterodimeric partner ErbB2 (Fig. 2a, b, Supplementary Fig. 2A–D)[51–53]. In fact, ErbB2, ErbB3, and the key ErbB3 effector pAkt are enriched in ruffles in $Nf2^{-/-}$ SCs compared to WT SCs, despite similar levels of total ErbB2, ErbB3, pAkt, and pERK (Supplementary Fig. 1C, 2A–E). Moreover, in contrast to WT SCs that restrict N-cadherin-based cell-cell contacts to bipolar cell tips, $Nf2^{-/-}$ SCs progressively make stable adhesive contacts around their entire periphery, from which cortical ruffles emerge (Fig. 2a, b). At late confluence $Nf2^{-/-}$ SCs form tightly packed adhesive aggregates that eventually lose attachment to the laminin-coated culture dish as spheroids (Fig. 2c, Supplementary Fig. 2F). In contrast, WT SCs do not adhere to one another, and instead undergo contact inhibition of locomotion (CIL), during which they redirect their movement after the collision until confluence when they round up as individual cells (Fig. 2d, e; Supplementary Fig. 2F, Supplementary Movie 1 and 2)[54]. Notably, WT and $Nf2^{-/-}$ SCs express similar levels of N-cadherin (Cdh2) and of Sox2, which can induce relocalization of N-cadherin to SC contacts, and little

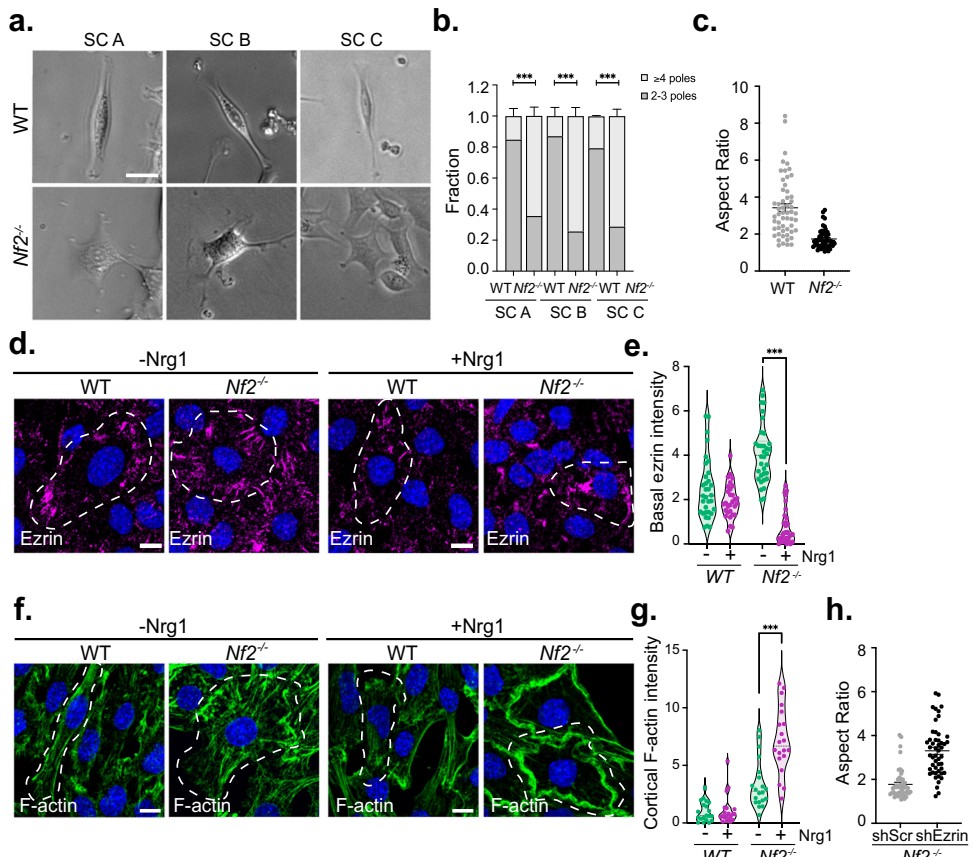

**Fig. 1 | Nf2-deficient Schwann cells exhibit enhanced multipolarity and exaggerated cytoskeletal responses to Nrg1 availability. a, b** Representative phase contrast images and quantitation of polarity in three independently generated $Nf2^{-/-}$ and control parental $Nf2^{flox/flox}$ (WT) SC lines. Data represent the fraction of cells containing 2–3 poles or ≥4 poles. $n = 123$ (WT SC A), $n = 420$ ($Nf2^{-/-}$ SC A), $n = 127$ (WT SC B), $n = 85$ ($Nf2^{-/-}$ SC B), $n = 377$ (WT SC C), $n = 633$ ($Nf2^{-/-}$ SC C). **c** Aspect ratio (length of long axis: length of short axis) in $Nf2^{-/-}$ and WT SC lines. $n = 55$ (WT) or $n = 50$ ($Nf2^{-/-}$) cells. **d** Confocal images depicting ezrin localization in WT and $Nf2^{-/-}$ SCs with and without Nrg1 stimulation (10 ng/ml Nrg1, 10 min). Dashed lines drawn according to F-actin distribution outline individual SCs to highlight their aspect ratio. **e** Violin plot depicting basal ezrin intensity in WT and $Nf2^{-/-}$ SCs with and without Nrg1. The area of ezrin-containing microvilli was measured at the basal cell surface (see methods) in thresholded images of ROIs at cell-cell boundaries. Data points represent the ratio of ezrin+ area to total cell area for $n = 33$ (WT, -Nrg1), $n = 30$ (WT, +Nrg1), $n = 34$ ($Nf2^{-/-}$, -Nrg1), or $n = 31$ ($Nf2^{-/-}$, +Nrg1) cells. **f** Confocal images depicting F-actin distribution in WT and $Nf2^{-/-}$ SCs with and without Nrg1 stimulation. Dashed lines are drawn as in **d**. **g** Violin plot depicting cortical F-actin intensity in WT and $Nf2^{-/-}$ SCs with and without Nrg1 stimulation, measured as the ratio of cortical F-actin area to total cell area in thresholded images of the cortical cell surface (see methods). Data points depict $n = 21$ (WT, -Nrg1), $n = 27$ (WT, +Nrg1), $n = 30$ ($Nf2^{-/-}$, -Nrg1), or $n = 30$ ($Nf2^{-/-}$, +Nrg1) cells. **h** Aspect ratio in $Nf2^{-/-}$ SCs infected with shSCR- or shEzrin-expressing lentiviruses. Data points represent $n = 50$ cells per condition; $N = 2$ independent experiments. All data are depicted as mean ± SEM and P values calculated with unpaired two-tailed Welch's $t$ test. ***$p < 0.0001$. Scale bars, 10 μm. Source data are provided as a Source Data file.

E-cadherin (Supplementary Fig. 2G, H)[53], suggesting that $Nf2$-deficiency does not drive increased adhesion by these mechanisms. Thus, the presence of Nrg1 $Nf2$-deficiency enables the progressive expansion of an adhesive apicojunctional ErbB signaling compartment.

Growth factor-stimulated cortical actin ruffles can trigger macropinocytosis, a form of extracellular nutrient engulfment that is a signature of $Nf2$-deficiency in other cell types[40, 46]. Indeed, in the presence of Nrg1, $Nf2^{-/-}$ but not WT SCs exhibit uptake of the macropinocytic cargo dextran-488 that is blocked by the macropinocytosis inhibitor 5-(N-ethyl-N-isopropyl) amiloride (EIPA) (Fig. 2f–h). Upon internalization, dextran-488 can be observed in large macropinosome-like vesicles adjacent to ruffling cell-cell contacts (Supplementary Fig. 3A). Murine $Nf2$-mutant schwannoma cells also exhibited striking Nrg1–stimulated ruffling and EIPA-sensitive dextran-488 uptake that was reversed by $Nf2$ re-expression or ezrin depletion (Supplementary Fig. 3B–J). Importantly, in vivo and ex vivo approaches revealed prominent EIPA-sensitive dextran-TMR uptake in schwannomas arising in the well-established $Postn$-$Cre;Nf2^{flox/flox}$ mouse model (Fig. 2i–k; see below)[55]. This observation could have immediate therapeutic ramifications as large macrotherapeutics largely gain entry into cells via

macropinocytosis[56]. As one example, extracellular vesicles (EVs) can be engineered to deliver many therapeutic cargoes, including nucleic acids for gene replacement[57–59]. We found that $Nf2^{-/-}$ schwannoma cells readily take up EVs in an EIPA-sensitive manner (Fig. 2l). These data suggest that cells in schwannomas can exhibit exaggerated cortical ruffling and tumor-selective uptake of macrotherapeutics in the presence of Nrg1, but when Nrg1 is not available, instead adopt a basally oriented cytoskeleton and exhibit neither ruffling nor macropinocytosis.

## $Nf2^{-/-}$ SCs develop auto/paracrine heterogeneity

In vivo, the unrestricted expansion of cell-cell contact, junctional Nrg1-signaling, and cortical ruffling exhibited by cultured $Nf2^{-/-}$ SCs would occur when intercellular contact is permitted and Nrg1 is available. However, cells within schwannomas do not retain axonal contact and must adapt to the loss of axonally provided Nrg1. We reasoned that $Nf2^{-/-}$ SCs might adapt by producing their own Nrg1, like other tumors that deploy autocrine ErbB ligand expression to induce cortical ruffling and macropinocytic scavenging[60]. We found that Nrg1 deprivation did yield modestly increased $Nrg1$ expression in both WT and $Nf2^{-/-}$ SCs

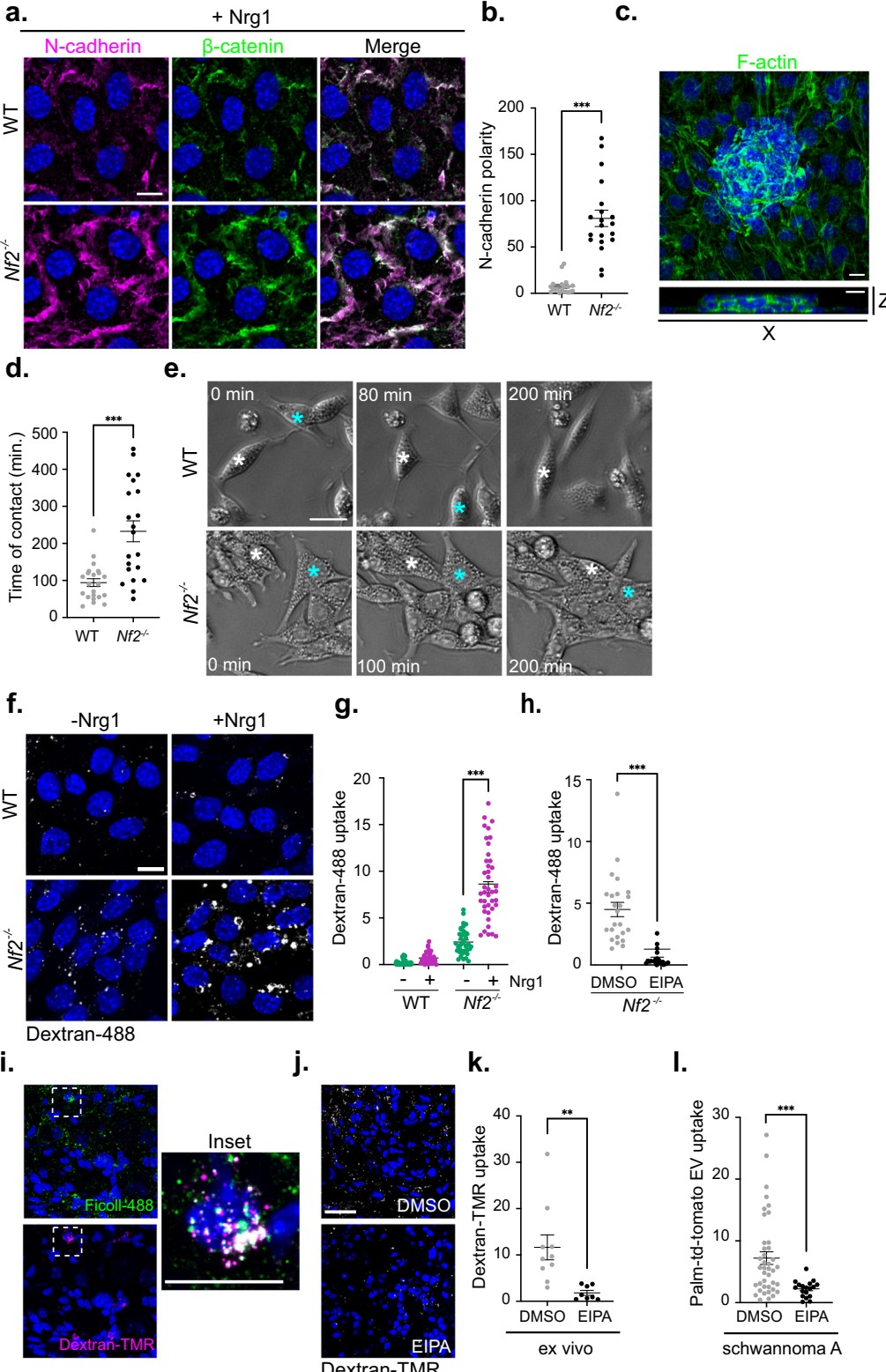

but caused a dramatic upregulation of the EGFR-specific ligands *Tgfa and Egf* specifically in *Nf2⁻/⁻* SCs (Fig. 3a, b). Surprisingly however, although EGF ligands trigger cortical ruffles and macropinocytosis in many cell types, including *Nf2⁻/⁻* liver epithelial cells[40,60], both EGF and TGFα suppress these phenotypes in *Nf2⁻/⁻* SCs, and instead enhance basal actin stress fibers and ezrin-containing basally oriented microvilli (Fig. 3c–e, Supplementary Fig. 4A; compare to Fig. 1d–g)[50]. In fact,

Nrg1−deprived *Nf2⁻/⁻* SCs also upregulated multiple components of the SC basal signaling platform, including laminin (*Lama2*), the laminin receptors α6 integrin (*Itga6*), and α-dystroglycan (*Dag1*), and collagen (*Col1a2*) consistent with the enhanced basal cytoskeletal organization seen under conditions of Nrg1−deprivation (Fig. 3f; Supplementary Fig. 4B)[14]. Both Nrg1−deprived WT and *Nf2⁻/⁻* SCs also upregulated *Egfr*, which is normally expressed at low levels in SCs but is often increased

**Fig. 2 | *Nf2⁻/⁻* SCs exhibit enhanced cell-cell contact and macropinocytosis in the presence of Nrg1. a** Confocal images depicting N-cadherin (magenta) and β-catenin (green) localization in Nrg1–stimulated WT and *Nf2⁻/⁻* SCs. **b** Quantitation of polarized N-cadherin intensity in Nrg1–stimulated WT and *Nf2⁻/⁻* SCs, measured as the ratio between N-cadherin area along the longest axis of a cell and N-cadherin area along the shortest axis of the cell. *n* = 20 junctions per condition. **c** *Top*, MIP of x-y section showing late confluent *Nf2⁻/⁻* SCs labeled for F-actin forming a tightly packed cluster of cells that loses contact with the laminin-coated coverslip. *Bottom*, vertical x-z section. **d**, **e** Duration of sustained contact after WT and *Nf2⁻/⁻* SCs collide (**d**) as calculated from time-lapse phase contrast images (**e**). *n* = 22 (WT) or *n* = 21 (*Nf2⁻/⁻*) cells. Asterisks in **e** mark two individual cells that come into contact. See Videos S1 and S2. **f** Uptake of the macropinocytosis cargo dextran-488 with or without Nrg1–stimulation (30 min) in WT and *Nf2⁻/⁻* SCs. **g** Quantitation of dextran-488 uptake. *n* = 37 (WT, -Nrg1), *n* = 38 (WT, +Nrg1), *n* = 37 (*Nf2⁻/⁻*, -Nrg1), or *n* = 40 (*Nf2⁻/⁻*, +Nrg1) cells. **h** Graph showing dextran-488 uptake in *Nf2⁻/⁻* SCs treated with vehicle (DMSO) or 50 μM EIPA. *n* = 24 (DMSO) or *n* = 21 (EIPA) cells. **i** Confocal images of macropinocytic uptake of FITC-Ficoll (green; tail vein injected) and dextran-488 (magenta; applied ex vivo) in a DRG lesion in a 12 month-old *Postn-Cre;Nf2^flox/flox* mouse. **j**, **k** Confocal images and quantitation of dextran-TMR uptake with and without EIPA treatment (0.5 mM) ex vivo in a DRG lesion. Scale bar, 30 μm. *n* = 10 (DMSO) or *n* = 8 (EIPA) cells. **l** Uptake of Palm-tdTomato-labeled EVs into *Nf2⁻/⁻* schwannoma cells after treatment with vehicle (DMSO) or 50 μM EIPA. *n* = 40 cells for DMSO, *n* = 18 cells for EIPA. All data are depicted as mean ± SEM. All *P* values were calculated with unpaired two-tailed Welch's *t* test. \*\*\**p* < 0.0001. Scale bars, 10 μm unless otherwise indicated. Source data are provided as a Source Data file.

in schwannomas, some of which respond clinically to pharmacologic EGFR inhibition (Supplementary Fig. 4C)[6,7,61]. Wild-type SCs neither upregulate EGF ligands nor ruffle in response to them (Fig. 3b; Supplementary Fig. 4D). Thus, *Nf2⁻/⁻* SCs may adapt to Nrg1–deprivation by upregulating an EGFR-associated basal polarity program.

In addition to Nrg1, peripheral nerves provide nutrients to adjacent SCs and vice versa. An important example may be the glutamate-glutamine cycle which is well-studied in the central nervous system[62,63]. In other types of tumors, glutamine depletion induces autocrine ruffling and macropinocytosis[60]. Indeed, we found that in *Nf2⁻/⁻* SCs glutamine depletion strongly induces ruffling and dextran-488 uptake that is blocked by the pan-ErbB inhibitor afatinib (ErbBi; Fig. 4a; Supplementary Fig. 5A, B). Consistent with the ability of Nrg1 but not TGFα or EGF to induce ruffling, the expression of *Nrg1 (type I)*, but not *Tgfa* is dramatically enhanced under these conditions (Fig. 4b; Supplementary Fig. 5C, E). Glutamine deprivation did not affect ErbB receptor levels or the levels of *Nrg2* or *Nrg3* (not detected) (Supplementary Fig. 5D, E). Conditioned medium from glutamine-deprived *Nf2⁻/⁻* SCs induced ruffling and EIPA- and ErbBi-sensitive dextran-488 uptake, while conditioned medium from Nrg1-deprived *Nf2⁻/⁻* SCs, like recombinant TGFα or EGF, enhanced basal structures (Fig. 4c–e, Supplementary Fig. 5F). Notably, *Nrg1* upregulation was completely rescued by the addition of exogenous glutamate (Fig. 4f). Altogether these data suggest that *Nf2*-deficient SCs can activate auto/paracrine programs of Nrg1-driven apicojunctional signaling and macropinocytic scavenging or EGF ligand-driven basal signaling in the absence of nutrients normally provided by axons.

## Heterogeneous mTOR activation and drug sensitivity

In normal SCs in vivo axonal Nrg1 drives the polarized activation of mTORC1, as measured by mTORC1-dependent levels of pS6[14]. Aberrant mTORC1 signaling has been implicated in schwannoma, and the pharmacologic mTORC1 pathway inhibitor everolimus (mTORC1i) reduced pS6 levels in mouse schwannoma models and in human schwannoma patients[36,64,65]. However, mTORC1i has exhibited only variable and cytostatic effects in human schwannoma patients thus far[36,37,64,65]. Our studies suggest that pS6 may be a specific biomarker of Nrg1 but not TGFα production in schwannoma. Indeed, pS6 levels are elevated in nutrient-starved Nrg1-expressing *Nf2⁻/⁻* but not WT SCs (Fig. 5a, b). Moreover, treatment of *Nf2⁻/⁻* SCs with conditioned medium from glutamine-deprived Nrg1-expressing *Nf2⁻/⁻* SCs dramatically stimulated pS6 while medium from Nrg1-deprived, TGFα-expressing *Nf2⁻/⁻* SCs dramatically lowered pS6 levels (Fig. 5c). Importantly, however, we found that high Nrg1-expressing, pS6+ *Nf2⁻/⁻* SCs are resistant to mTORC1i relative to Nrg1-deprived, *Tgfa*-expressing pS6- *Nf2⁻/⁻* SCs, and mTORC1i has no impact on *Nrg1* expression and instead reduces *Tgfa* expression (Fig. 5d, e, Supplementary Fig. 6). On the other hand, ErbBi blocks *Nrg1* but not *Tgfa* expression (Fig. 5f). These results suggest that the upregulation of Nrg1 by glutamine-deprived *Nf2⁻/⁻* SCs occurs in a mTORC1-independent fashion, which could explain the cytostatic and heterogeneous clinical efficacy of mTORC1i

in human patients, as well as the 'rebound' tumor growth that is often seen upon cessation of mTORC1i treatment and could be triggered by a reservoir of accumulated Nrg1[64,65]. We note that while pS6 has been shown to be a readout of mTORC1 in *Nf2⁻/⁻* SCs and in mouse and human schwannomas, mTOR circuitry is complex and may be subject to context-dependent feedback and additional inputs[36,64-66]. Indeed, our data have important ramifications for designing and interpreting therapeutic strategies for schwannoma and beg an investigation of this circuitry via biomarker analysis of tumor tissue.

## Self-generated heterogeneity

Collectively, our observations suggest that when deprived of axonal signals, *Nf2⁻/⁻* SCs can adopt different programs of coordinated ligand production and cytoskeletal polarity, featuring high *Nrg1* expression, high pS6 levels and apicojunctional polarity dominated by cortical actin ruffling and macropinocytosis, or high *Tgfa* expression, low pS6 and basal polarity dominated by basal actin stress fibers and ECM receptors (Fig. 6a, b). The progressive ligand- and ECM-associated features of *Nf2⁻/⁻* SC 'states' could enable self-generated patterns of heterogeneity within schwannomas. To test this possibility, we developed a three-dimensional (3D) model in which single *Nf2⁻/⁻* SCs are embedded in inert polyethylene glycol (PEG) hydrogels in the absence of the polarity cues imposed by a laminin-coated surface in standard two-dimensional SC culture[67]. After 4 days of culture in the presence of Nrg1, single *Nf2⁻/⁻* SCs expanded to form solid spheres in which pS6+ cells were distributed throughout (Fig. 6c, d, Supplementary Fig. 7). In contrast, when deprived of Nrg1, pS6+ cells became restricted to the outer perimeter of the spheres in a striking example of spatially patterned heterogeneity (Fig. 6c, d). Consistent with the idea that it is triggered by auto/paracrine Nrg1, pS6 levels in these spheres were blocked by the pan-ErbBi afatinib but not the EGFR-specific inhibitor erlotinib, which actually interfered with the spatial patterning of pS6 itself (Fig. 6e, f). Moreover, this peripherally restricted distribution of pS6+ cells was eliminated by exogenous TGFα or an external polarizing cue provided by functionalizing the PEG hydrogels with integrin-binding RGD peptides (Fig. 6g). In fact, the combination of exogenous TGFα and RGD peptides suppressed pS6 activity altogether and instead strongly induced pNDRG, an effector of basal integrin signaling in normal SCs (Fig. 6g–i)[14]. Thus *Nf2⁻/⁻* SCs can self-generate heterogeneous cellular patterns according to biomarkers of the phenotypic states that we identified in standard cell culture and those patterns are influenced by both auto/paracrine ligands and polarizing external cues.

## Quantitative analysis of self-generated heterogeneity in schwannoma

Next, we looked for evidence of self-generated heterogeneity in schwannoma tissue. We developed a quantitative pipeline to study schwannoma heterogeneity by immunofluorescence (IF)-based antibody staining of formalin-fixed paraffin-embedded (FFPE) sections, whole-slide scanning of the entire tumor section, and digital image

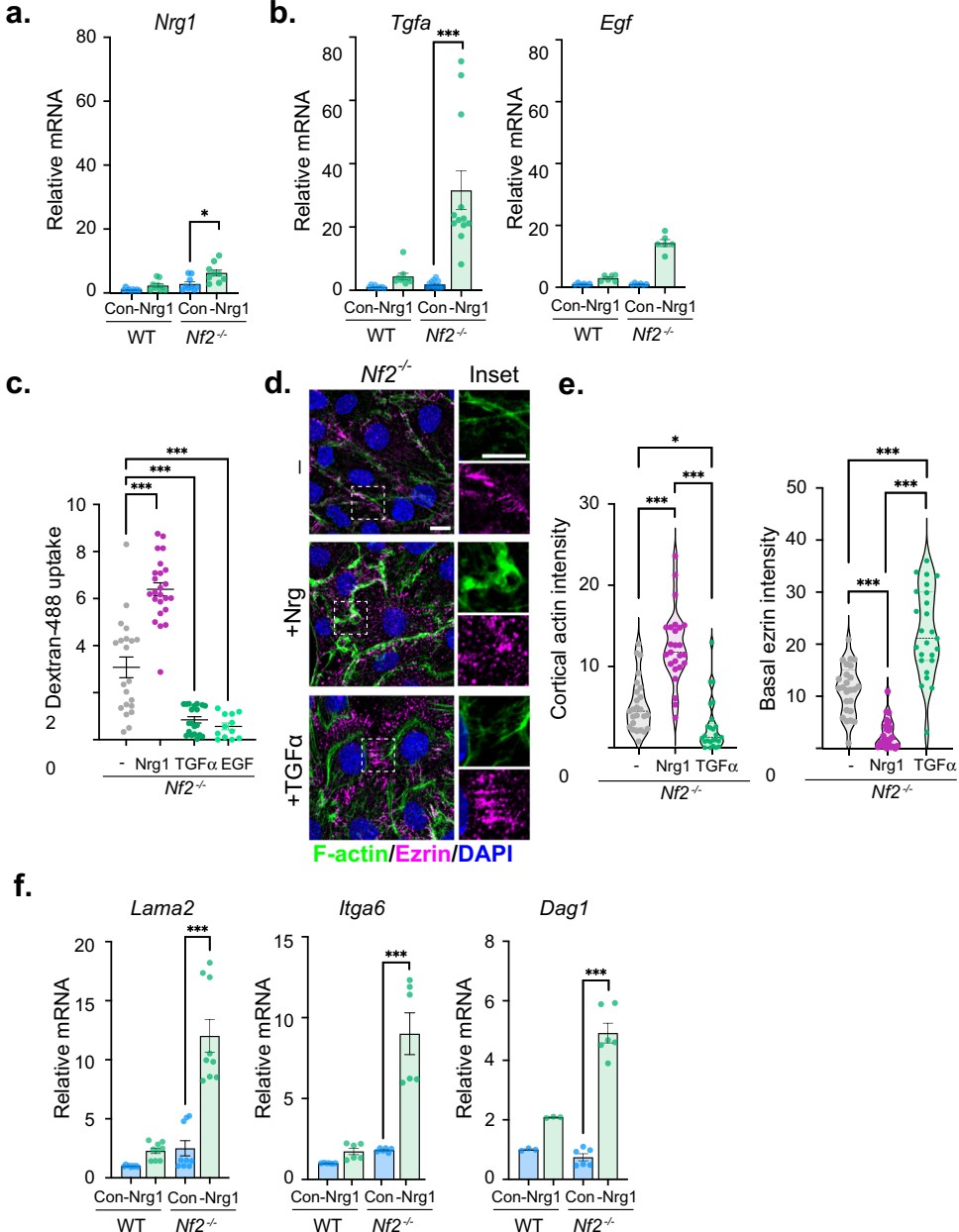

**Fig. 3 | *Nf2*⁻/⁻ SCs upregulate basal polarity components upon Nrg1−deprivation. a** Levels of *Nrg1* mRNA in WT and *Nf2*⁻/⁻ SCs grown in complete medium (control; Con) or deprived of Nrg1 for 24 h. Data are presented as mean ± SEM relative to WT mRNA levels. *n* = 3 replicates. **b** Levels of *Tgfa* and *Egf* mRNA in WT and *Nf2*⁻/⁻ SCs grown under complete medium or deprived of Nrg1 for 24 h. Data are presented as mean ± SEM relative to WT mRNA levels for *n* = 3, *N* = 3 independent experiments (WT, *Tgfa*); *N* = 4 independent experiments (*Nf2*⁻/⁻; *Tgfa*); *N* = 2 independent experiments (WT, *Nf2*⁻/⁻, *Egf*). **c** Quantitation of macropinocytic dextran-488 uptake in *Nf2*⁻/⁻ SCs starved of Nrg1 overnight and stimulated with 10 ng/ml Nrg1, TGFα, or EGF. *n* = 18 (starved), *n* = 20 (Nrg1), *n* = 16 (TGFα), or *n* = 8 (EGF) cells

and lines represent mean ± SEM. **d** Confocal images depicting the distribution of F-actin (green) and ezrin (magenta) in *Nf2*⁻/⁻ SCs starved of Nrg1 overnight and stimulated with 10 ng/ml Nrg1 or TGFα. Scale bars, 10 μm. **e** Violin plots depicting cortical actin intensity and basal ezrin intensity in cells in **d**. *n* = 25 cells per condition. **f** Levels of *Lama2*, *Itga6*, and *Dag1* mRNA in WT and *Nf2*⁻/⁻ SCs in complete medium or after 24 h Nrg1−deprivation. Data are presented as mean ± SEM relative to WT mRNA levels for *n* = 3 replicates. *P* values were calculated with unpaired two-tailed Welch's *t* test (**a**, **f**) or one-way ANOVA with Tukey's multiple comparisons test (**c**, **e**). *\**p* = 0.0132 (**a**), 0.0162 (**e**), \*\*\**p* < 0.0001. Source data are provided as a Source Data file.

analysis using HALO™ software. Quantitative analysis of the distribution of NRG1, pS6 and pNDRG in tissue sections from four different human vestibular schwannomas (VS) revealed that each exhibited striking intra- and intertumoral heterogeneity (Fig. 7a, b, Supplementary Fig. 8A). Visual examination of the scanned tumor tissue revealed that the tumors exhibited large regions that were positive and negative for each biomarker (Fig. 7a, Supplementary Fig. 8A, B). Spatial analysis confirmed that, as predicted by our in vitro studies, there was strong regional concordance between Nrg1 and pS6, and discordance

between NRG1/pS6 and the basal biomarker pNDRG (Fig. 7c−e, Supplementary Fig. 8B, C). Interestingly there was often a striking demarcation between NRG1+/pS6+ and pNDRG+ regions within a given tumor (Fig. 7e).

To understand how regional patterns of heterogeneity develop in schwannoma, we analyzed *Postn-Cre;Nf2^flox/flox* mice[55]. We found that in these mice schwannomas develop synchronously in each of 15 pairs of dorsal root ganglia (DRG) between 3 and 6 months of age (Fig. 8a), providing a unique opportunity to study the evolution of both intra-

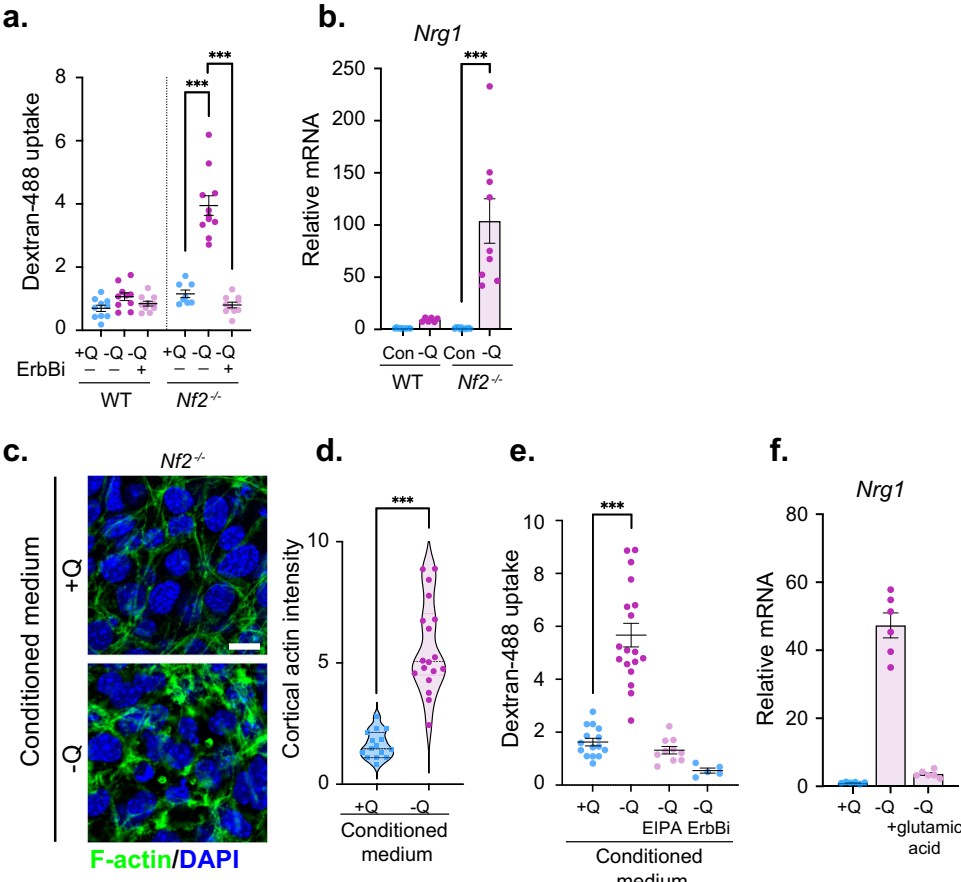

**Fig. 4 | *Nf2^-/-* SCs enact a distinct feedforward autocrine Nrg1–signaling program upon nutrient deprivation. a** Macropinocytic dextran-488 uptake in WT and *Nf2^-/-* SCs in response to 24 h glutamine deprivation with and without treatment with 2 μM ErbBi. Data points are graphed for *n* = 10 (WT, all conditions), *n* = 8 (*Nf2^-/-*, +Q), *n* = 11 (*Nf2^-/-*, −Q), or *n* = 10 (*Nf2^-/-*, −Q/ErbBi) cells and lines represent mean ± SEM. **b** Levels of *Nrg1* mRNA in WT and *Nf2^-/-* SCs grown for 24 h in complete medium or glutamine-deprived medium (−Q). Data are presented as mean ± SEM relative to WT mRNA levels for *n* = 3 replicates. **c** Confocal images of F-actin depicting cortical ruffling in *Nf2^-/-* SCs stimulated for 10 min with conditioned medium from glutamine-deprived (−Q) but not glutamine replete (+Q) *Nf2^-/-* SCs. Scale bar, 10 μm. **d** Violin plot depicting cortical actin from cells in **c**. Data points are

shown for *n* = 15 (+Q) or *n* = 18 (−Q) cells. **e** Dextran-488 uptake in *Nf2^-/-* SCs stimulated for 30 min with conditioned medium from glutamine replete (+Q) *Nf2^-/-* cells or from glutamine-deprived (−Q) *Nf2^-/-* cells treated with vehicle (DMSO), 50 μM EIPA, or 2 μM ErbBi. Data points represent *n* = 15 (+Q), *n* = 18 (−Q), *n* = 10 (-Q, EIPA), or *n* = 5 (-Q, ErbBi) cells and lines represent mean ± SEM. **f** Levels of *Nrg1* mRNA in glutamine-deprived (−Q) *Nf2^-/-* SCs with or without the addition of 10 mM L-glutamic acid. Bars represent mean ± SEM relative to +Q, DMSO mRNA levels for *n* = 3 replicates. *N* = 2 independent experiments. *P* values were calculated with unpaired two-tailed Welch's *t* test (**d**) or one-way ANOVA with Tukey's multiple comparisons test (**a**, **b**, **e**). ****p* < 0.0001. Source data are provided as a Source Data file.

and intertumoral heterogeneity. We first examined Nrg1 and TGFα in DRG lesions in a 6 month-old *Postn-Cre;Nf2^flox/flox* mouse at single cell resolution, applying a Random Forest machine learning algorithm to exclude large DRG soma, which nevertheless served as important geographical landmarks (Fig. 8b). This analysis revealed intra-lesion heterogeneity for both Nrg1 and TGFα at this early timepoint (Fig. 8b, c, Supplementary Fig. 9A). Importantly, simultaneous analysis of the distribution of Nrg1 and TGFα protein and mRNA by fluorescent in situ hybridization-immunofluorescence (FISH-IF) revealed excellent concordance, suggesting that the patterns of ligand distribution are mostly due to autocrine expression and that diffusion of the secreted ligands in the tumor tissue has minimal impact on these patterns (Supplementary Fig. 9B, C). Co-staining of mRNA-identified cell subpopulations revealed broad anti-correlation between *Nrg1*- and *Tgfa*-expressing cells and inter-lesional heterogeneity for these populations across DRG, while also confirming the existence and heterogeneous distribution of cells expressing neither or both *Nrg1* and *Tgfa* (Fig. 8d, e, Supplementary Fig. 9D, E). Notably, analysis by both IF and FISH-IF revealed a spatial bias of Nrg1+ cells in proximity to neuronal soma; pS6 also exhibited a soma-proximal bias, while pNDRG1 localized farther away from soma (Fig. 8f, g; Supplementary Fig. 9F). Finally,

zooming in revealed striking spatial patterns of both protein and mRNA within tumor substructures. Cellular 'whorls', which are prominent features of early schwannoma lesions in this model and in human NF2 patients, exhibited concentric patterns of biomarker distribution, with Nrg1/*Nrg1*+ and pS6+ cells often concentrated in the central portion and even 'bullseye', and *Tgfa*+ and pNDRG+ cells at the whorl periphery (Fig. 8h, Supplementary Fig. 9C, G). Altogether these observations highlight the concept of self-generated heterogeneity as an early step in tumorigenesis and the value of this model and approach to understanding it.

## Discussion

Heterogeneity has emerged as a central challenge to the successful treatment of many human tumor types and until recently was thought to be driven entirely by underlying genetic variation and consequent tumor:microenvironment interactions[68,69]. Recent studies, particularly of diffuse gliomas of the central nervous system, have uncovered important epigenetic contributions to tumor heterogeneity but how they arise are poorly understood[70–72]. Heterogeneity is also a major issue in schwannoma despite the benign nature and unique genetic homogeneity of these common tumors; it is currently not possible to

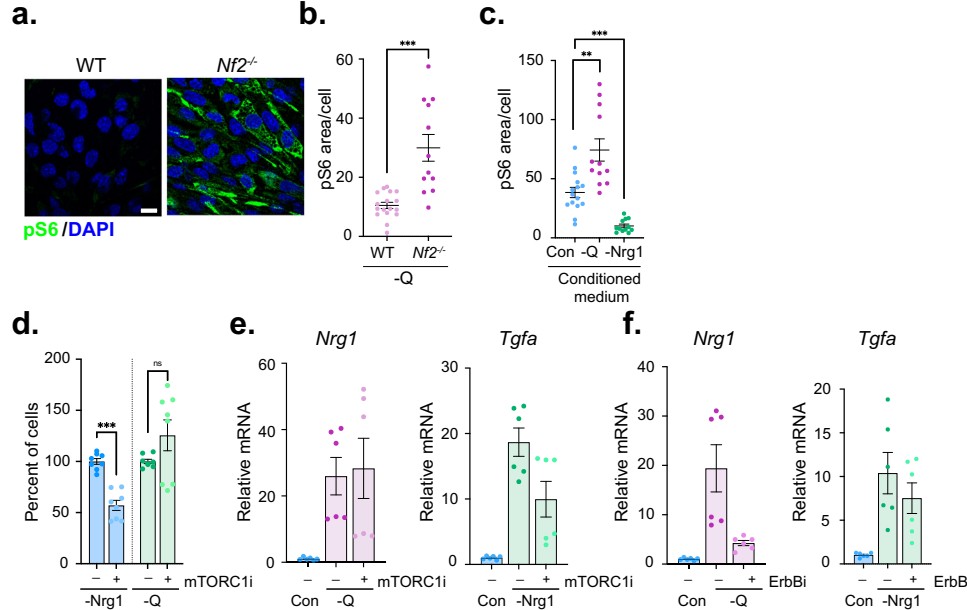

**Fig. 5 | Heterogeneous mTOR activation and drug sensitivity. a** Confocal images depicting the dramatic induction of pS6 (green) in glutamine-deprived *Nf2*⁻/⁻ SCs but not WT SCs. Scale bar, 10 μm. **b** Quantitation of pS6 levels in cells in **a** as measured in thresholded confocal images as the ratio between pS6 area and total cell area. Data points are graphed for *n* = 12 (WT) or *n* = 17 (*Nf2*⁻/⁻) cells per condition and lines represent mean ± SEM. **c** Quantitation of pS6 levels in *Nf2*⁻/⁻ SCs treated with conditioned medium from glutamine- (−Q) or Nrg1−deprived *Nf2*⁻/⁻ SCs. Data points are shown for *n* = 15 (Con.) or *n* = 12 (-Q, -Nrg1) cells per condition, and lines represent mean ± SEM. **d** Quantitation of cell viability of *Nf2*⁻/⁻ SCs deprived of Nrg1 or glutamine upon treatment with mTORC1i (100 nM, 72 h). Bars represent mean ± SEM percentage of cells compared to vehicle treatment for *n* = 8 samples per condition over *N* = 3 independent experiments. **e** *Left*, levels of *Nrg1* mRNA in glutamine-deprived *Nf2*⁻/⁻ SCs treated with vehicle (DMSO) or 100 nM mTORC1i. *Right*, levels of *Tgfa* mRNA in Nrg1−deprived *Nf2*⁻/⁻ SCs treated with vehicle (DMSO) or 100 nM mTORC1i. Data are presented as mean ± SEM relative to control, DMSO mRNA levels for *n* = 3 replicates. *N* = 2 independent experiments. **f** Levels of *Tgfa* mRNA in Nrg-deprived *Nf2*⁻/⁻ SCs treated with vehicle (DMSO) or 2 μM ErbBi. Data are presented as mean ± SEM relative to control, DMSO mRNA levels for *n* = 3 replicates. *N* = 2 independent experiments. *P* values were calculated with unpaired two-tailed Welch's *t* test (**b**, **d**) or one way ANOVA with Tukey's multiple comparisons test (**c**). ns = not significant, ***p* < 0.0001. Source data are provided as a Source Data file.

predict schwannoma growth rate, impact on nerve function, or therapeutic response, rendering clinical management difficult. While understanding the underpinnings of schwannoma heterogeneity is essential for improving therapeutic options for these patients, the genetic simplicity of schwannomas also provides a unique window into the broadly important contribution of intrinsic mechanisms to tumor heterogeneity.

Our studies suggest that the ability of *Nf2*⁻/⁻ SCs to variably enact distinct programs of ErbB ligand production and cytoskeletal polarity upon loss of axonally provided nutrients is a key intrinsic driver of schwannoma heterogeneity. This would be consistent with the variable activation of multiple signaling pathways in *Nf2*⁻/⁻ SCs, including mTORC1, mTORC2, and YAP/TAZ[22, 30, 36]. Indeed, while the mTORC1 effector pS6 is induced only when auto/paracrine Nrg1 is available, known SC YAP/TAZ targets *Itga6* and *Dag1* are elevated only upon Nrg1−deprivation, which hyperactivates the basal polarity program, consistent with the well-known activation of YAP/TAZ in response to cell-ECM attachment (Fig. 3f)[73, 74]. Variability in the contribution of these states and their distinct auto/paracrine signaling and metabolic programs to each tumor could underlie the heterogeneous clinical behavior and therapeutic response of schwannomas. For example, metabolic dysfunction in glial cells is known to cause neurodegeneration and our studies suggest that heterogeneous metabolic programming is a feature of schwannoma that could differentially impact adjacent nerves[15, 21]. Differential signaling triggered by the two polarity states could also explain the heterogeneous therapeutic responsiveness seen for mTORC1- and EGFR-inhibiting drugs in schwannoma patients[6, 8]. Our discovery that pS6 is a biomarker of Nrg1 expression but *not* of mTORC1i sensitivity in cultured *Nf2*⁻/⁻ SCs, together with the observation that mTORC1i does not block Nrg1 expression itself while ErbBi does, speaks to the complex clinically relevant circuitry that

drives the development of these genetically simple tumors. It will be important to further dissect the ErbB-mTOR signaling circuitry and determine whether schwannoma cells are able to adaptively modify it in the face of pharmacologic signaling blockades in vivo. It is interesting to note that auto/paracrine Nrg1 expression has been recently reported in *NF2*-mutant meningioma, but in this case is blocked by mTORi, revealing cell-type specific signaling circuitry[75].

A deeper mechanistic understanding of how unstable polarity triggers differential auto/paracrine ligand expression as schwannomas initiate and evolve requires a way to monitor it spatially in 3D and in association with nerves. In vivo, intrinsic SC polarity is spatially informed by contact between the apicojunctional surface and the axon, and the two polarized SC surfaces are balanced by unknown mechanisms of feedback[14]. Unstable polarity may disable this feedback mechanism, yielding progressive loss of axonal contact and the acquisition of cell-cell or basal lamina contact and associated auto/paracrine ligand production in an essentially stochastic manner. On the other hand, schwannomas predominantly form on sensory nerves, of which there are many subtypes; the striking sensitivity of *Nf2*⁻/⁻ SCs to glutamine-glutamate availability suggests that schwannoma initiation and evolution may be biased by the nerves that they are in contact with. Overall, the self-generating design principle we propose here likely also applies to genetically complex tumors, including epithelial cancers that exhibit variable polarized cell architecture, and gliomas, in which unbiased single-cell approaches have uncovered tumor cell subpopulations of distinct epigenetic states that may be related to those we identified through hypothesis-driven investigation[71, 72].

The unique availability of a mouse model in which multiple *Nf2*-mutant schwannomas develop synchronously in anatomically defined locations provides an opportunity to quantitatively and deeply study schwannoma heterogeneity. By imaging and analyzing multiple lesions

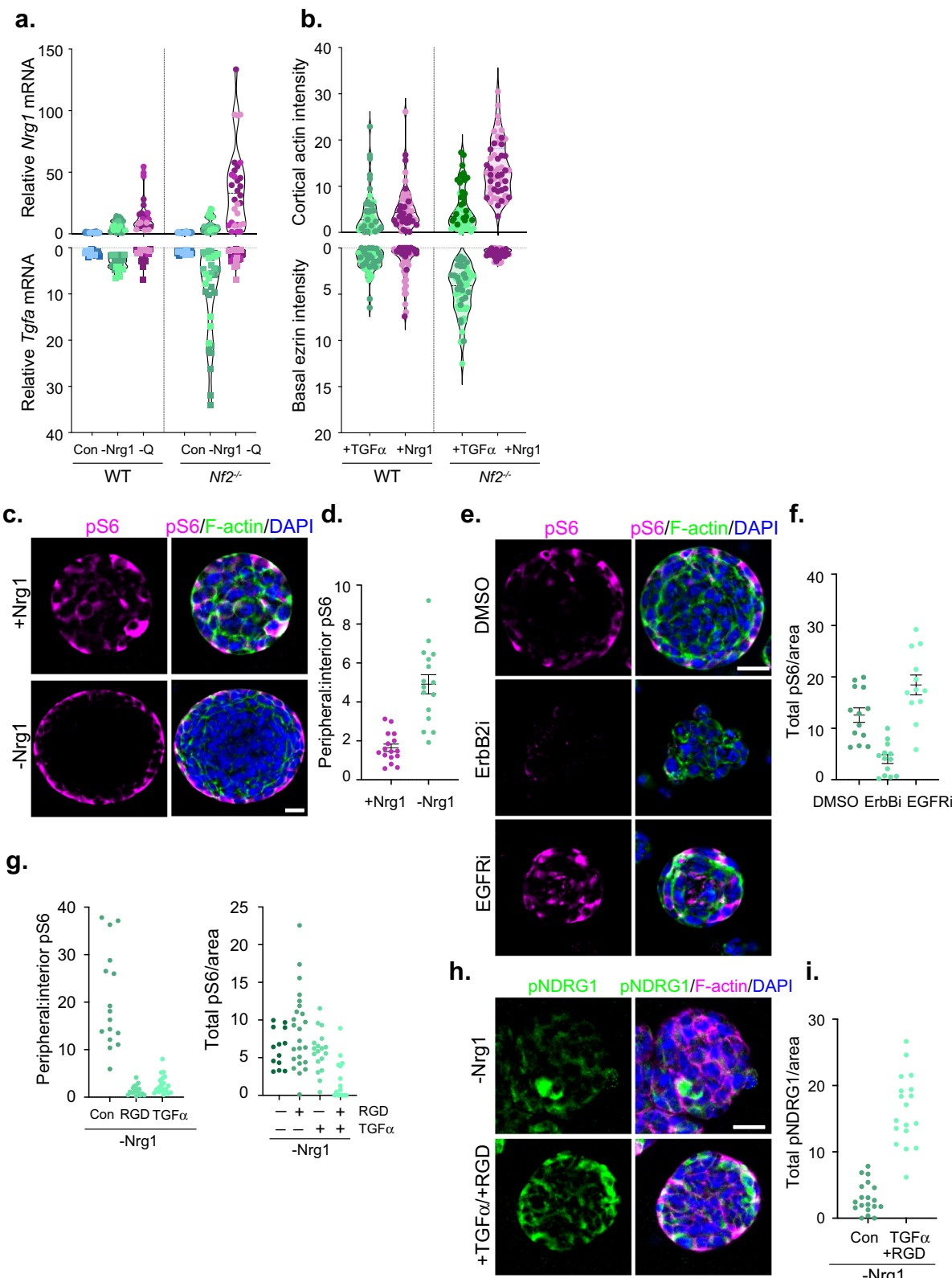

in a single mouse, we captured the power of this model to both confirm and extend aspects of our in vitro studies. In addition to validating *Nrg1*/Nrg1 and *Tgfa*/Tgfα mRNA and protein as anticorrelated biomarkers of schwannoma heterogeneity, we found a spatial bias of Nrg1–expressing cells within lesions in proximity to DRG soma, a source of Nrg1 that could drive the feedforward Nrg1 expression that we see in vitro. Notably, it has been reported that neuronal Nrg1 levels

are reduced by *NF2* haploinsufficiency in some nerves[45]; moreover, Nrg1 is also regulated by cleavage and can trigger phosphorylation of the Nf2 protein merlin, which could influence neighboring Nf2-expressing SCs[47]. These mechanisms could further contribute to schwannoma complexity in NF2 patients. Our studies set the stage for a comprehensive analysis of how intrinsic tumor heterogeneity evolves over time, and how stromal, endothelial, and immune cell types are

**Fig. 6 | Self-generated heterogeneity. a** Violin plots depicting *Nrg1* and *Tgfa* mRNA levels in three independently generated *Nf2*⁻/⁻ and WT SC lines under steady-state conditions (blue) or deprived of Nrg1 (green) or glutamine (magenta) for 24 h. Each data point represents fold change in mRNA relative to steady-state conditions of the same genotype. **b** Violin plots depicting polarized cytoskeletal organization in *Nf2*⁻/⁻ and WT SCs stimulated with TGFα (green) or Nrg1 (magenta) for 10 min. Each data point represents actin or ezrin intensity in a single cell. **a, b** Each SC line is represented by a different color shade (light, medium, dark). **c, d** Confocal images and quantitation depicting pS6 distribution (magenta) in single cell-derived spheres of *Nf2*⁻/⁻ SCs cultured in polyethylene glycol (PEG) gels in the presence and absence of Nrg1, measured as the ratio of peripheral (outermost layer) to inner cells. *n* = 16 spheres/condition. **e, f** Confocal images and quantitation of pS6 in

DMSO-, EGFRi- or pan-ErbBi-treated *Nf2*⁻/⁻ PEG spheres. Data represents pS6 per total sphere area for *n* = 13 (DMSO, ErbBi) or *n* = 12 (EGFRi) spheres. *N* = 2 independent experiments. **g** Quantitation of peripheral (*left*) and total (*right*) pS6 in *Nf2*⁻/⁻ spheres grown in PEG gels with and without RGD peptides (0.5 mM) or TGFα (10 ng/ml). *n* = 14 (Con.), *n* = 15 (RGD), or *n* = 22 (TGFα) spheres for peripheral distribution and *n* = 13 (Con.), *n* = 18 (RGD), n = 24 (TGFα), or *n* = 15 (RGD, TGFα) spheres for total pS6. *N* = 2 independent experiments. **h, i** Confocal images and quantitation showing pNDRG1 distribution and total levels in *Nf2*⁻/⁻ PEG spheres in the absence (-Nrg1) or presence of both TGFα and RGD peptides. *n* = 18 (Con.) or *n* = 19 (RGD, TGFα) spheres. *N* = 2 independent experiments. All data depicted as mean ± SEM. Scale bars, 20 μm. Source data are provided as a Source Data file.

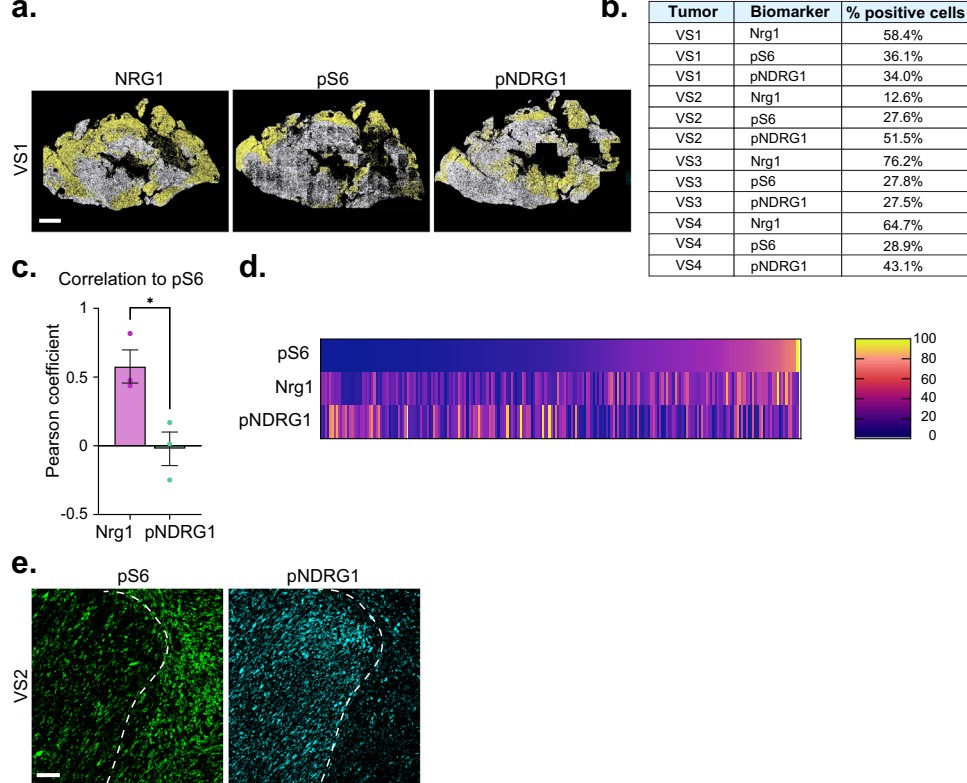

**Fig. 7 | Quantitative analysis of self-generated heterogeneity in human schwannoma tissue. a** Segmentation analysis of NRG1, pS6, or pNDRG1 staining across entire serial sections of a representative human vestibular schwannoma (VS1). Scale bar, 1 mm. **b** HALO-mediated quantitation of percentage of NRG1, pS6, or pNDRG1-positive cells across entire tumor sections of four different human vestibular schwannomas (VS1, 2, 3, 4). **c** Summary of correlation analysis of NRG1+ and pNDRG1+ cells with pS6 as measured by Pearson correlation coefficient. Bars

represent mean values ± SEM for *N* = 3 tumors. *P* value was calculated with unpaired two-tailed Welch's *t* test *\*p* = 0.0251. **d** Heatmap showing levels of NRG1 and pNDRG1 relative to pS6 levels across 3 VS tumor sections (*n* = 16 ROIs in 5 20× fields of view per tumor). **e** Confocal images showing the distribution of pS6 and pNDRG1 in VS2. Dotted line depicts the boundary between pS6+ and pNDRG1+ regions of the tumor. Images representative of *N* = 4 tumors. Scale bar, 50 μm. Source data are provided as a Source Data file.

spatially recruited to expanding schwannomas. Additional biomarkers can be multiplexed to create multivariate heterogeneity indices for schwannoma that could be mapped to both tumor behavior (ie growth rate or nerve dysfunction) and therapeutic response, with the ultimate goals of predicting tumor behavior, guiding treatment strategy, and improving therapeutic options for sporadic schwannoma and particularly NF2 patients, for whom few options exist.

## Methods

### Mice
The mouse schwannoma cell line was derived from a tumor dissected from a *PO-CreB;Nf2^flox/flox* mouse[76,77]. The genetically engineered mouse model *Postn-Cre;Nf2^flox/flox* of NF2 schwannomas was generated by breeding *Postn-Cre* and *Nf2^flox* mice[76,78]. All mouse strains were

maintained on FVB/N genetic background. All animal care and experimentation were performed with the approval of the UCLA Institutional Animal Care and Use Committees under protocol number 2019-042. Mice were housed under standard conditions: 12 hours of light/12 hours of dark; ambient temperature range 20–26 °C; ambient humidity range 30–70%. Mice were monitored twice a week until dissection at 6 or 12 months of age. Mice were euthanized by $CO_2$ inhalation. Per UCLA IACUC guidelines, all mice were euthanized before or when tumors interfered with the animal's health and well-being, or when the body condition score (BCS) decreased.

### Cell culture and reagents
Primary murine SCs were isolated from sciatic nerves of *Nf2^flox/flox* adult mice and purified by magnetic sorting as previously described[48,76]

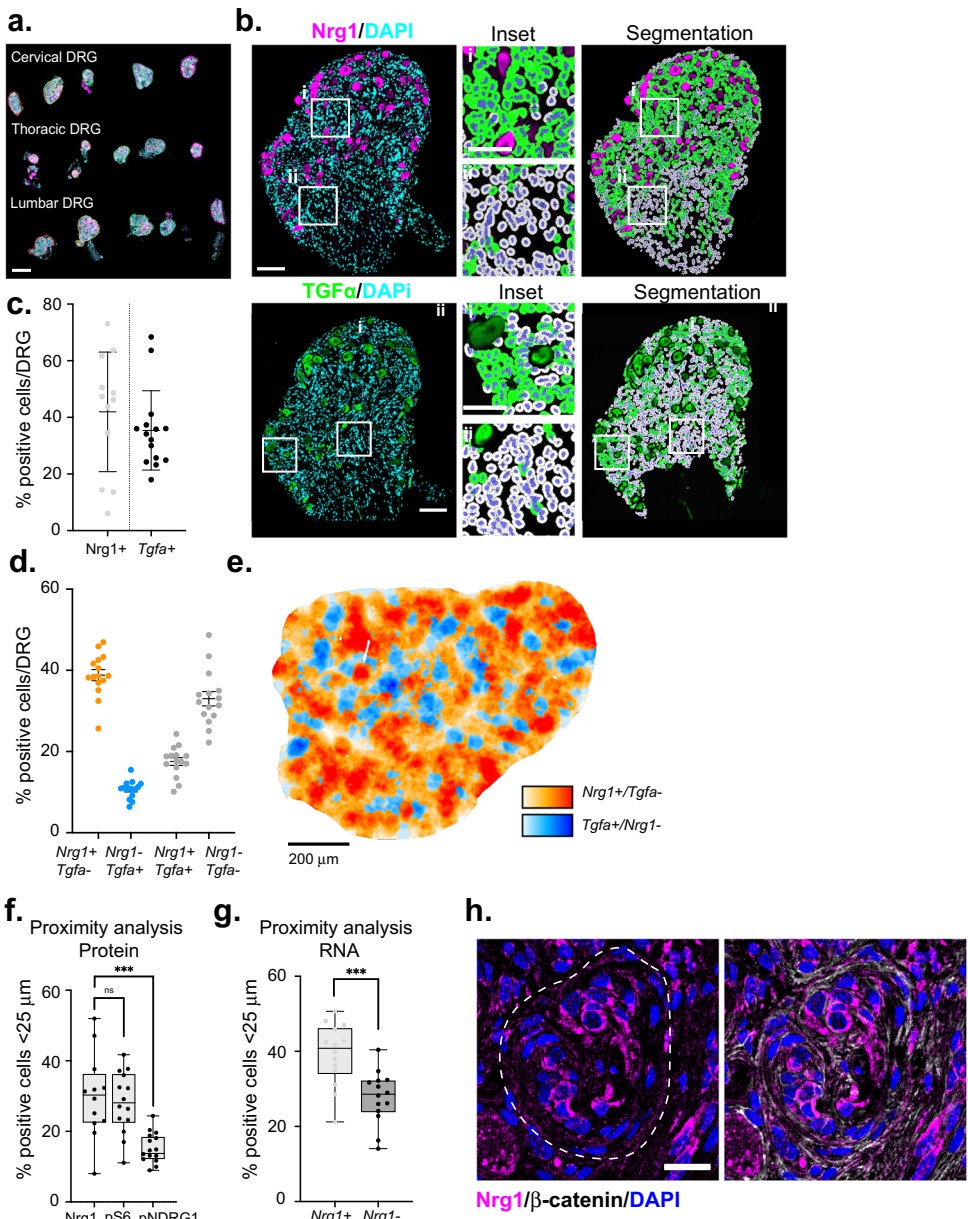

**Fig. 8 | Quantitative analysis of heterogeneity in schwannomas arising in *Postn-Cre;Nf2*ˡᵒˣ/ˡᵒˣ DRG. a** Representative whole-slide scan of DRG array from a 6 month-old *Postn-Cre;Nf2*ᶠˡᵒˣ/ᶠˡᵒˣ mouse stained for TGFα (magenta) and DAPI (cyan). Scale bar, 1 mm. **b** Top left, a DRG lesion from a 6 month-old *Postn-Cre;Nf2*ᶠˡᵒˣ/ᶠˡᵒˣ mouse stained for Nrg1 (magenta) and DAPI (cyan); *top right*, overlaid HALO-mediated segmentation filter depicting Nrg1+ (green) and Nrg1− (gray) cells, and neuronal soma (magenta). *Insets*, Nrg1 expression close to (i) and far from (ii) DRG soma. Bottom left, image of TGFα (green) and DAPI (cyan) staining; bottom right, overlaid HALO-mediated segmentation filter depicting TGFα+ (green) and TGFα− (gray) cells, and neuronal soma (green). Insets, TGFα expression close to (i) and far from (ii) DRG soma. Scale bars, 100 μm. **c** Percentage of Nrg1+ or TGFα+ cells per DRG in 6-month-old lesions. (*n* = 12, Nrg1; *n* = 15, TGFα). **d** Quantitation of *Nrg1*+ and *Tgfa*+ mRNA-expressing cells per DRG in a 12-month-old *Postn-Cre;Nf2*ˡᵒˣ/ˡᵒˣ mouse by FISH. Data points represent the percentage of positive cells per DRG (*n* = 15).

**e** Pseudocolored representation of the spatial distribution of *Nrg1*+ and *Tgfa*+ mRNA-expressing cells across an individual DRG lesion. Scale bar, 200 μm.
**f** Proximity analysis measuring the percentage of Nrg1+, pS6+, or pNDRG1+ cells within 25 μm of DRG soma in 6 month-old lesions. Each data point represents the percentage of positive cells per DRG (*n* = 12, Nrg1; *n* = 14, pS6; *n* = 17, pNDRG1)
**g** Proximity analysis measuring the percentage of *Nrg1* mRNA-expressing versus -non-expressing cells within 25 μm of DRG soma. Each data point represents the percentage of positive cells per DRG (*n* = 14). **h** Left, Confocal image of the bulls eye-like distribution of Nrg1+ cells within a whorl in a DRG lesion. *Right*, counterstaining for β-catenin reveals additional cellular intra-whorl architecture. Scale bar, 20 μm. Data are presented as mean ± SEM. Box plots extend from 25th to 75th percentiles, with center line representing the median and whiskers representing minimum to maximum. ***$p < 0.0001$, unpaired two-tailed Welch's *t* test. Source data are provided as a Source Data file.

(available from authors upon request). SCs were routinely cultured in N2 medium (DMEM/F12-HAM, 1x N2 supplement (Thermo Fisher), 50 μg/ml gentamicin (Thermo Fisher), 2 μM forskolin (Calbiochem) and 10 ng/ml Nrg1 (HRG-beta-1 EGF domain) (R&D Systems) on poly-L-lysine (50 μg/ml, Millipore Sigma) and mouse laminin (10 μg/ml, Thermo Fisher) coated plates and incubated at 37 °C and 7.5% $CO_2$. *Nf2*

was deleted from *Nf2*ᶠˡᵒˣ/ᶠˡᵒˣ SCs via adenovirus infection with Cre-recombinase (Ad5-CMV-Cre). Tumoral SCs from mouse schwannomas were isolated by dispase/collagenase dissociation as described[48] and cultured in DMEM supplemented with 10% fetal bovine serum (FBS), 1x N2 supplement, 50 μg/ml gentamicin, 2 μM forskolin, and 10 ng/ml Nrg1 on poly-L-lysine and laminin-coated plates at 37 °C and 7.5% $CO_2$

(available from authors upon request). Adenovirus infection was used for $Nf2$ re-expression (Ad5-CMV-Nf2$^{WT}$). 293 A cells for adenovirus production (240085, Agilent), and 293 T cells for lentivirus production (CRL-3216, ATCC) were cultured in 10% FBS-DMEM with 1% penicillin/streptomycin (Thermo Fisher) and incubated at 37 °C and 5% CO$_2$. For 3D cell culture $Nf2^{-/-}$ SCs were trypsinized, centrifuged at 200 × $g$ for 5 minutes (min), and resuspended in complete N2 medium. Cells were then added at a concentration of 5.0 ×10$^5$ cells/ml to a mixture containing 2 mM SLO-dextran polymer and 2 mM PEG non-cell degradable cross-linker in TrueGel 3D buffer, pH 7.2 (TrueGel3D Hydrogel Kit, Millipore Sigma) and plated in an 8 well chamber slide. Where indicated an RGD peptide (sequence - Acetyl-Cys-Doa-Doa-Gly-Arg- Gly-Asp-Ser-Pro-NH$_2$; Doa = 8-amino-3,6-dioxaoctanoic acid, Millipore Sigma) was added to the cell suspension at a concentration of 0.5 mM. Hydrogels were allowed to gel for 50 min at room temperature before overlaying with N2 medium. Following incubation at 37 °C and 7.5% CO$_2$ for 48 hours (h), complete N2 medium was either replenished or replaced with Nrg1−free N2 medium or Nrg1−free, TGFα-containing N2 medium and incubated for an additional 72 h.

### Growth factors and reagents

Nrg1 (HRG-beta-1 EGF domain) was used at 10 ng/ml; All other growth factors were obtained from Peprotech. EGF was used at 10 ng/ml; TGFα was used at 10 ng/ml . Drug pretreatment was as follows: EIPA (Millipore Sigma), 50 μM, 60 min; afatanib (Selleck), 2 μM, 24 h; erlotinib (Selleck), 2 μM, 24 h; everolimus (Selleck), 100 nM, 24 h. These doses were maintained throughout the experiment.

### Plasmids and shRNA constructs

The $Nf2^{WT}$ expression construct was generated by PCR amplification of the mouse $Nf2$ coding region and cloned into a pAdCMV vector as described[39]. The shRNA constructs targeting mouse ezrin (5′-ATTTCCTTGTTATAATCTCCG-3′) in a pLKO-puro.1 vector from GE Healthcare and described in ref. [27]. The control (shScr; (5′-CA GTCGCGTTTGCGACTGG-3′) in a pLKO-puro.1 vector was provided by Marianne James (MGH, Boston)[79].

### Virus production and infection

$Nf2^{WT}$-expressing and Cre-recombinase adenoviruses were generated using the AdEasy system (Agilent) as described in ref. [39]. Cells were infected 24 h before the start of the experiment to induce $Nf2$ gene expression. An empty adenoviral vector was used as a control (EV). shRNA-expressing lentiviruses were generated by co-transfecting 293 T cells with pLKO-puro.1 vectors and the packaging vectors ΔVPR and VSVG (FuGENE, Promega). Viruses were harvested 24-48 h post transfection. shScr or shEzrin-expressing lentiviruses were stably expressed in LDCs and selected in 4 μg/ml puromycin.

### Antibodies

The following primary antibodies were used: anti-ezrin mouse monoclonal antibody (mAb) (1:500; clone 3C12, #MA5-13862, Invitrogen); anti-p75 NTR rabbit mAb (1:1000; clone D8A8, #4201, Cell Signaling Technology); anti-N-cadherin mouse mAb (1:500; clone 32, #610921, BD Biosciences); anti-beta-catenin mouse mAb (1:500; clone 14, #610154, BD Biosciences); anti-pAkt (S473) rabbit mAb (IF: 1:100, WB: 1:1000; clone D9E, #4060, Cell Signaling Technology); anti-Akt rabbit mAb (1:1000; clone 11E7, #4685, Cell Signaling Technology); anti-phospho p44/42 (ERK1/2) rabbit mAb (1:1000; clone D13.14.4E, #4370, Cell Signaling Technology); anti-p44/42 (ERK1/2) rabbit pAb (1:1000; #9102, Cell Signaling Technology); anti-ErbB2 rabbit pAb (IF: 1:100; A0485, Agilent), anti-ErbB3 mouse mAb (IF: 1:100, WB: 1:1000; clone RTJ2, #MA1-860, Invitrogen); anti-E-cadherin mouse mAb (1:1000; clone 36, #610182, BD Biosciences); anti-actin mouse mAb (1:1000;

clone AC-40, #A4700, Millipore Sigma); anti-merlin rabbit mAb (1:100; clone D1D8, #6995, Cell Signaling Technology); anti-pS6 (S235/236) rabbit pAb (1:100; #2211, Cell Signaling Technology); anti-Nrg1 rabbit pAb (1:100; #ab191139, Abcam); anti-TGFa rabbit pAb (1:100; #ab9585, Abcam); anti-pNDRG1 rabbit mAb (1:100; clone D98G11, #5482, Cell Signaling Technology); and anti-Neurofilament L rabbit mAb (1:100; clone C28E10, #2837, Cell Signaling Technology). Alexa Fluor 647-phalloidin or Rhodamine-phalloidin (1:500; Thermo Fisher) was used to label F-actin. The following secondary antibodies were used for IF at a dilution of 1:500: goat anti-rabbit IgG (H + L) highly cross-adsorbed secondary antibody, Alexa Fluor 488 (#A-11034, Invitrogen); goat anti-rabbit IgG (H + L) highly cross-adsorbed secondary antibody, Alexa Fluor 555 (#A-21428, Invitrogen); goat anti-rabbit IgG (H + L) highly cross-adsorbed secondary antibody, Alexa Fluor 647 (#A-21244, Invitrogen); goat anti-mouse IgG (H + L) highly cross-adsorbed secondary antibody, Alexa Fluor 488 (#A-11001, Invitrogen); goat anti-mouse IgG (H + L) highly cross-adsorbed secondary antibody, Alexa Fluor 555 (#A-21422, Invitrogen); and goat anti-mouse IgG (H + L) highly cross-adsorbed secondary antibody, Alexa Fluor 647 (#A-21235, Invitrogen). For western blotting, ECL anti-Rabbit IgG, horseradish peroxidase-linked whole antibody (from donkey) (#GENA934, Millipore Sigma) and ECL anti-mouse IgG, horseradish peroxidase-linked whole antibody (from sheep) (#GENA931, Millipore Sigma) were used at a dilution of 1:5000. DAPI (1:1000, Invitrogen) was used to label nuclei.

### Time-lapse imaging

60,000 cells per condition were plated and allowed to adhere for 15 min before washing away unbound cells. Images of regions of interest were captured at ×20 magnification at time intervals of 5 min between cycles for 24 h. Cells were imaged on a fully automated Nikon TiE microscope (Micro Device Instruments) equipped with a biochamber heated at 37 °C and 5% CO$_2$ as described in ref. [80].

### Immunofluorescence microscopy

Cells for immunofluorescence were plated on glass coverslips coated with poly-L-lysine and laminin 24−48 h before staining. Early confluent cells were fixed and stained when ~80% of the coverslip was covered with cells. For late confluent cells, cells were incubated for 24-48 h after reaching early confluence prior to fixation. Cells were fixed in 4% paraformaldehyde in PBS for 15 min at room temperature, then permeabilized in 0.2% Triton X-100 for 10 min. Primary and secondary antibodies were diluted in PBS with 1% BSA and incubated for 1 h at room temperature. Alexa Fluor-Phalloidin was added with secondary antibodies. Coverslips were mounted with Prolong Gold antifade mountant (Thermo Fisher). Cells were imaged with an inverted laser scanning confocal microscope (LSM710; Carl Zeiss) equipped with a 63x oil immersion objective (Plan Apochromat NA 1.4; Carl Zeiss). DAPI was excited with a 405-nm laser line of a diode laser. Alexa Fluor 488, Alexa Fluor 555, and Rhodamine probes were excited with the 488-nm or 514-nm laser line of an argon laser. Texas-Red fluorescent probes were excited with the 561-nm laser line of a helium-neon laser. Alexa Fluor 647 probes were excited with the 633-nm laser line of a helium-neo laser. Images were acquired as single images or z-stacks in sequential mode using ZEN Black software (version 2.1, Carl Zeiss).

### Western blotting

Cells were lysed in RIPA buffer (50 mM Tris, pH 7.4, 1% Triton X-100, 1% SDS, 0.5% sodium deoxycholate, 150 mM NaCl, 1 mM EDTA, 1 mM EGTA, and protease inhibitors). Cell debris was cleared by centrifugation (18,000 × $g$ for 10 min at 4 °C) and equal amounts of protein were separated by SDS-PAGE, transferred to PVDF membranes, and immunoblotted with primary antibody overnight at 4 °C in 5% milk or 5% BSA (phospho-specific antibody).

## Macropinocytosis assays

For in vitro assay, cells were plated on poly-L-lysine and laminin-coated glass coverslips and incubated for 24-48 h until close to confluent. Cells were starved of serum and/or Nrg1 overnight in either complete growth medium or glutamine-free medium. Dextran-488, (Oregon Green 488; 70,000 da, anionic, lysine fixable) or was added to cells along with growth factor when appropriate (0.5 mg/ml) and incubated for 30 minutes at 37 °C. Cells were rinsed 3 times with cold PBS and fixed with 4% paraformaldehyde. For the in vivo/ex vivo assay, *Postn-Cre;Nf2*$^{flox/flox}$ mice were injected with 250 mg/kg FITC-Ficoll (70,000 Da, Millipore Sigma 51731-1 G) and dorsal root ganglia (DRG) were dissected 24 h later. Dissected tumors were incubated overnight in N2 growth medium without Nrg1 and then with 1 mg/mL dextran-TMR (10,000 Da, Molecular Probes D1817) for 1 h in complete N2 growth medium with Nrg.1 For the EIPA ex vivo assay, DRG were dissected from *Postn-Cre;Nf2*$^{flox/flox}$ mice, incubated 24 h in N2 growth medium without Nrg with DMSO or EIPA (500 μM) pretreatment. DRG were then incubated for 30 min in N2 growth medium with Nrg1 and 1 mg/mL dextran-TMR. After treatments, DRG were directly frozen in OCT blocks and sectioned at 8 μm.

## EV production and uptake

EVs were harvested as previously described[81] from the conditioned medium of ~$1.0 \times 10^8$ EO771-LMB cells expressing tandem dimer Tomato (tdTomato) fused to the NH2-terminus to a palmitoylation signal (Palm-tdTomato)(kindly provided by Drs. Xandra Breakfield and Shannon Stott). Medium was collected and centrifuged at $300 \times g$ for 10 min and at $2000 g$ for 10 min. Supernatants were filtered (0.8 μm; EMD Millipore) and ultracentrifuged at $100,000 \times g$ for 90 min at 4 °C (Optima L-90K Ultracentrifuge, Beckman Coulter). Pelleted EVs were resuspended in 200 μl PBS. EVs were added to cells grown on poly-L-lysine and laminin-coated glass coverslips at ~$10^6$ per 0.7 cm$^2$ and incubated for 4 h at 37 °C. Cells were then fixed and stained with DAPI.

## Nutrient deprivation and conditioned medium experiments

Cells were plated in complete N2 medium and allowed to grow for 24-48 h until they reached near confluence. Complete N2 medium was replaced with glutamine-free or Nrg1–free medium, as indicated, and incubated for 24 h. For experiments in schwannoma cells serum was also removed from the medium. For glutamate rescue experiments, 10 mM L-glutamic acid (Millipore Sigma) was added to glutamine-free medium and incubated with cells for 24 h. For conditioned medium experiments, medium was harvested from *Nf2*$^{-/-}$ SCs grown on 60 mm plates in complete growth medium, glutamine-free medium, or Nrg1–free medium for 72 h. Medium was centrifuged at $400 g$ for 5 min and filtered through a 0.45 μm filter. 100 μl of conditioned medium was added to each coverslip containing *Nf2*$^{-/-}$ SCs for the indicated time.

## Quantitative RT-PCR

Total cellular RNA was extracted using TRIzol (Thermo Fisher) and reverse transcribed with MMLV-RT (Promega) using oligo-dT primers. Fast Start Universal SYBR Green mix (Millipore Sigma) was used to amplify 0.5 μl of the RT reaction in a 25 μl total reaction volume. Triplicate samples were run on a Light Cycler 480 system (Roche Applied Science) with cycling conditions of denaturation for 15 seconds at 95 °C, annealing for 1 minute at 60 °C, and extension at 60 °C, 45 cycles. Expression of GAPDH was used as an internal reference gene.

Primer sequences are as follows: *Gapdh* forward (5′-AGGTC GGTGTGAACGGATTTG-3′), reverse (5′-TGTAGACCATGTAGTTGAGGT CA-3′); *Nrg1* forward (5′-TCATCTTCTAGCGAGATGTCTG-3′), reverse (5′-CAGACATCTCGCTAGAAGATGA-3′); *Nrg1, Type I* forward (5′-GGG AAGGGCAAGAAGAAGG-3′), reverse (5′-TTTCACACCGAAGCACGA GC-3′); *Nrg1, TypeIII* forward (5′-ACTCAGCCACAAACAACAGAAAC 3′), reverse (5′-GAAGCACTCGCCTCCATT-3′); *Tgfa* forward (5′-CACT CTGGGTACGTGGGTG-3′), reverse (5′-CACAGGTGATAATGAGGACA GC-3′); *Egf* forward (5′-AGCATCTCTCGGATTGACCCA-3′), reverse (5′-CCTGTCCCGTTAAGGAAAACTCT-3′); *Lama2* forward (5′-TCCCAAG CGCATCAACAGAG-3′), reverse (5′-CAGTACATCTCGGGTCCTTTTT C-3′); *Itga6* forward (5′-TGCAGAGGGCGAACAGAAC-3′), reverse (5′-CG TGCTGCCGTTTCTCATATC-3′); *Dag1* forward (5-'CAGACGGTACGGC TGTTGTC-3', reverse (5′-AGTGTAGCCAAGACGGTAAGG-3'); and *Egfr* forward (5′-GCCATCTGGGCCAAAGATACC-3′), reverse (5′-GTCTTCG CATGAATAGGCCAAT-3′).

## Immunohistochemistry and tissue staining

Immunofluorescence staining was performed on frozen sections dissected from *Postn-Cre;Nf2*$^{flox/flox}$ mice using a standard protocol: after air drying the frozen sections, they were fixed in formaldehyde 3.7% for 15 min, washed in PBS 3 times, incubated in a blocking/permeabilization buffer (PBS + 5%NGS + 0.3%Triton) for 1 h. Sections were then incubated with primary antibodies overnight at 4 °C, washed with PBS 3 times and incubated with secondary antibodies for 1 h at room temperature. Nuclear staining was performed with Hoechst 33258 (0.2 μg/μL) and sections were mounted with Fluorescence Mounting Medium (Dako S3023). For staining of macropinocytosis assays, slides were washed quickly in PBS before fixation to reduce nonspecific background and then incubated with Hoechst before coverslip mounting[82]. Images were acquired as single-plan in sequential mode using a SP8 Light-Sheet confocal microscope and the LAS X software (Leica). The human samples were collected and analyzed under the David Geffen School of Medicine at UCLA IRB protocol 10-000655. Standard protocols were used for immunofluorescent staining of formalin-fixed paraffin-embedded human and mouse tissue. Briefly, tissues were dewaxed in xylene and rehydrated, followed by antigen retrieval in citrate buffer by incubation for 20 min at 95 °C. Sections were then blocked in 5% goat serum, 1% BSA in TBST for 1 h and then incubated with primary antibodies overnight at 4 °C. Sections were then washed, incubated with secondary antibodies for 1 h at room temperature, and incubated with DAPI to label nuclei prior to mounting on glass slides with Prolong Gold mounting media. Slides were scanned on a Vectra 3 whole-slide scanning microscope (Akoya Biosciences), viewed using Phenochart software, and spectrally unmixed using InForm software.

## Fluorescent in situ hybridization-Immunofluorescence (RNAscope) and image analysis

Dorsal root ganglion were dissected from a 12 month-old *Postn-Cre;Nf2*$^{flox/flox}$ mouse. Formalin-fixed paraffin-embedded tissue sections (3.5 μm) were stained following the Integrated Co-Detection Workflow combining in situ hybridization using the RNAscope Multiplex Fluorescent v2 assay (ACD, Advanced Cell Diagnostics) and immunofluorescence. After preparation of tissues according to the manufacturer's recommendations (Protocols 323100-USM and MK 51-150/Rev B), sections were steamed for target retrieval (15 min), and incubated with primary antibodies against NRG1 (ab191139, Abcam), TGFa (ab9585, Abcam) or Neurofilament L (NF-L, #2837, Cell Signaling Technology) overnight at +4 °C. Protease Plus was applied to the slides (30 minutes) and hybridization was performed using mouse Nrg1 (Cat# 418181, ACD) and Tgfa (Cat# 435251-C2, ACD) probes or RNAscope 3-plex Positive and Negative Control Probes. Opal dyes 520, 570 (FP1487001KT and FP1488001KT Akoya Biosciences) were used for signal development of the RNA probes. Then, NRG1 and TGFa primary antibodies were detected with HRP-conjugated antibody (ARR1001KT, Akoya Biosciences) and TSA-based amplification with Opal dye 690 (FP1497001KT, Akoya Biosciences). NF-L primary antibody was detected with an Alexa Fluor Plus 647-conjugated goat anti-rabbit IgG (H + L) secondary antibody (A32733, Invitrogen). Tissues were counterstained with DAPI and whole-slide images were obtained using a Vectra Polaris slide scanner with a 40x objective (Akoya Biosciences) and unmixed using InForm software (Akoya Biosciences). Image

analysis was performed using the HALO software v3.4 (Indica Labs) with module FISH-IF V2.1.5. For each staining, a classifier was created to exclude the neurons cell bodies from the analysis of mRNA probes. Nrg1 and Tgfa mRNA expression was measured using the semi-quantitative ACD scoring. Briefly, the number of dots/cell was assessed using the HALO FISH-IF module and cells categorized following ACD recommendations as Class 0 (0 dots/cell), Class 1 (1–3 dots/cell), Class 2 (4-9 dots/cell), Class 3 (10-15 dots/cell) or Class 4 (>15 dots/cell). Proximity analysis was performed with HALO between specific cell subpopulations and the neuron cell bodies, identified by the NF-L IF staining. Density heatmaps were generated using the same scale for the four cell subpopulations (0 to 10 cells per 25 μm radius). Heatmaps combining two subpopulations were generated by exporting single-color tinted density heatmaps and merging them using the "darker color" layer blending mode in Photoshop (Adobe).

## Image analysis

ImageJ software (version 2.0, National Institutes of Health) was used for all image processing and analysis. The displayed images were produced from single confocal slices or maximum intensity projections (MIP) of z-stack images. Background was removed with rolling ball background subtraction. Lookup tables were applied to produce final images. Cortical actin and basal ezrin area were defined using the approach described in ref. [83]. Briefly, in z-stack images stained with phalloidin to label F-actin and DAPI to label nuclei, the nuclear mid-point was determined and the F-actin labeled area above was defined as cortical actin and the ezrin labeled region below was defined as basal ezrin area. For ease of quantitation, a single confocal image from the top (cortical) or bottom (basal) region of the cell was used to measure cortical actin or basal ezrin in each image. Each channel was background subtracted and thresholding was used to generate a binary image. The thresholded mask was then used to generate pixel area measurements of the percentage of cortical actin area/total cell area. Total cell area was calculated by measuring the area of a region of interest (ROI) drawn around the periphery of each cell using the polygon tool. Fluorescence intensity was measured by applying a threshold mask to a ROI and calculating the area of positive signal to the total surface area of the cell, field of view, or region of interest. For cell polarity measurements, cell poles were either counted by hand in time-lapse images or the aspect ratio (defined as the length of the major axis of the cell divided by the length of the minor axis of the cell) was calculated using the shape descriptor plugin in ImageJ. Dextran or EV uptake was measured using the analyze particles function to determine the ratio of total particle area to total cell area in threshold images. For quantification of the ex vivo macropinocytosis inhibition by EIPA, analysis was performed with 5 single-plane confocal images per conditions, processed as described in ref. [82]. To quantify pS6 peripheral distribution in 3D spheres, the polygon tool was used to draw a ROI around only the outer layer of cells in a sphere and pS6 levels were measured in thresholded images of the ROI (peripheral). The same process was used to measure pS6 levels in the interior cells in ROIs containing the interior layers of cells. pS6 levels were graphed as the ratio of peripheral to interior pS6+ area relative to total area of the sphere. Whole-slide scans of DRG lesions from *Postn-Cre;Nf2^flox/flox* mice were imported into HALO™ digital image analysis software version 3.4 (Indica Labs). We then developed a digital image analysis algorithm using the HALO™ multiplex IHC algorithm to achieve single-cell segmentation of nucleus and cytoplasm and detect Nrg1 and TGFα at a single-cell resolution in DRG lesions. Cell recognition and nuclear segmentation were trained and optimized in three randomly selected regions from each section. Thresholds for positive staining for Nrg1 and TGFα were set using 3 randomly selected regions close to and far from DRG soma. A supervised machine learning algorithm (Random Forest classifier) was trained on three randomly selected regions to recognize tumor cells and exclude DRG soma. For spatial analysis, the HALO™ proximity analysis module was used to measure the percentage of positive cells within a 25 μm radius of DRG soma identified using the classifier algorithm.

## Statistics and reproducibility

Data from all analyses were imported into Prism 9 for plotting graphs and statistical analysis. The unpaired two-tailed Welch's t test was used to compare differences between the two groups. One-way ANOVA with Tukey's multiple comparisons test was used to compare data across more than two groups. All data are representative of at least three independent experiments unless otherwise specified.

## Reporting summary

Further information on research design is available in the Nature Portfolio Reporting Summary linked to this article.

## Data availability

The data that support the findings of this study are available in the article or the supplementary information files. Source data is available in the source data file provided with this paper. Source data are provided with this paper.

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

## Acknowledgements

We thank past and present members of the McClatchey lab for valuable discussions, particularly Christian Davidson for intellectual input in the early stages of this project; Berent Aldikacti for EO771-LMB cells and expertise in EV isolation; João Paulo Oliveira-Costa for advice on HALO-based image analysis; Daniel Irimia and Xiao Wang for assistance with time-lapse imaging; MGH Cancer Center/Molecular Pathology Confocal Core for access to confocal microscopy equipment. Confocal laser scanning microscopy was also performed at the Advanced Light Microscopy/Spectroscopy Laboratory and the Leica Microsystems Center of Excellence at the California NanoSystems Institute at UCLA with funding support from NIH Shared Instrumentation Grant S10OD025017 and NSF Major Research Instrumentation grant CHE-0722519. This work was supported by the U.S. Army Medical Research and Development Command, through the Neurofibromatosis Research Program under Award Nos. W81XWH-16-1-0086 (M.G.), W81XWH-19-1-0156 (A.I.M.), W81XWH-21-1-0446 (A.I.M.), W81XWH-21-1-0448 (M.G.). Opinions, interpretations, conclusions, and recommendations are those of the authors and are not necessarily endorsed by the Department of Defense. This work was also supported by Drug Discovery Initiative Registered Reports Awards from the Children's Tumor Foundation (2018-05-005, A.I.M. and 2020-05-004, J.V.), and an MGH American Cancer Society Institutional Research Grant (C.C.M.).

## Author contributions

The study was conceived and designed by C.C.M., J.V., M.G., and A.I.M. In vitro experiments were carried out by C.C.M., C.H.L., E.F., and E.W. In vivo experiments were carried out by J.V. Whole-slide scanning and HALO image analysis were carried out by C.C.M, J.V., E.F., and S.L.S. Data were analyzed and interpreted by C.C.M, J.V., C.H.L., M.G., and A.I.M. The project was supervised by A.I.M and M.G. Drafting of the manuscript and preparation of figures was completed by C.C.M, J.V., and A.I.M. All authors read and commented on the manuscript.

## Competing interests

The authors declare no competing interests.
