## [Peer Review File · Nature Communications]

Cellular mechanisms of heterogeneity in *NF2*-mutant schwannomaREVIEWER COMMENTS

Reviewer #1 (Remarks to the Author):

This report by two established leaders in the NF2 field focuses on understanding how heterogeneity develops in Nf2-mutant schwannomas using mouse modeling. The experiments are nicely designed, adequately powered, and appropriately interpreted.

Intrinsic polarity. Growing Schwann cells on laminin does not remove contextual cues, since laminin may provide instructive signals for polarity. This conclusion that this phenomenon is cell intrinsic should be independently validated in other contexts in vitro or using other methods.

Neuregulin-1. The choice of Nrg1 was not made clear. Work from other laboratories have shown that other neuregulin family members may also be potent regulators of Schwann cell growth and motility (reviewed in Stonecypher JNEN 2006). Is it possible that other Nrg family members account for the observed heterogeneity? Similarly, prior work from the Herrlich group revealed that merlin controls CD44 ectodomain cleavage (Mol Cancer Res 2015). Was this examined, since active (cleaved NRG1) is not regulated by merlin and is predominantly involved in cell differentiation? Lastly, Fernandez-Valle showed that NRG stimulate merlin phosphorylation (Oncogene 2008).

ErbB2. How do the authors reconcile differences in their findings and those reported by Morrison and colleagues (Brain 2014)?

Mechanistic insights. It is necessary to demonstrate cause and effect relationships between merlin and ErbB function/signaling. This is particularly relevant to dissecting the mechanism underlying merlin regulation of pinocytosis.

Autocrine heterogeneity. In this section, the authors present a series of observations, but do not provide conclusive etiologic evidence for any one of the mechanisms as being responsible for the phenotypes seen in Nf2-deficient Schwann cells.

mTOR dependence. A recent study also demonstrated that neuregulin-ErbB3 autocrine signaling in NF2-deficient tumor cells is controlled by mTOR (J Biol Chem 2021). It is not clear why the authors conclude that pS6 is a specific biomarker of autocrine Nrg1 signaling.

Heterogeneity in tumors. It would be helpful to demonstrate that Nrg1+ and TGFa+ cells have a similar pattern in human tumors. In addition, since both proteins are secreted, how confident are the investigators that they are capturing cell expression? RNAScope would be a more definitive method to demonstrate which cells are making these paracrine factors. It is also not clear in Figure 7 which cells in the tumor are expressing these factors.

Reviewer #2 (Remarks to the Author):

The goal of this study was to characterize the heterogeneity seen in Nf2-/- Schwann cells and determine the underlying molecular mechanisms of this heterogeneity. The authors characterized the phenotype of Nf2-/- Schwann cells and found them to have increased multipolarity, increased actin polymerization, increased cell-cell contacts, anchorage independence, increased dextran uptake (proxy for macropinocytosis), and increased autocrine signaling. They suggested that the autocrine signaling could set up some kind of stochastic process whereby cells become polarized to distinct phenotypes. However, I found that the focus on heterogeneity is distracting and confusing. The relationship between heterogeneity, multipolarity, increased actin polymerization, increased cell-cell contacts, etc. was not articulated or demonstrated. I feel it would be more straightforward for the authors to state the goal of the study was to simply understand the molecular mechanisms of schwannoma pathology.

- I'm not sure that I accept the claim that "An understanding of the molecular basis of

schwannoma heterogeneity is essential for the development of successful non-surgical therapies.” (Lines 51-53). If this were the case, then you would need to provide evidence from previous work showing that some phenotypes are treatable while others are not. Rather, is it that a lack of successful therapies is simply due to insufficient understanding of the molecular mechanisms of disease?

- General methodology questions:

- o Why use RNAi when you could use a CRISPR gene disruption system?

- o I do not have expertise in Schwann cell culture and cannot comment on the experimental model system of isolating cells from spontaneously generated mouse tumors in KO animals. However, as study of cellular heterogeneity seems especially sensitive to cellular context, differentiation, and so forth, it would be helpful if the authors provided justification or validation for their model system. In the area of cell biology that I work in, we would expect dramatically different phenotypes in cells isolated from different areas of the body, from mice of different ages and so forth. Can you more clearly state how this work is novel in that regard and how your experimental set-up is justified to study this question?

- Figure 1: This figure demonstrates the increased multipolarity, change in cellular morphology and increased cortical actin in the Nf2^{-/-} cells, but it doesn't show exceptional heterogeneity. It would help to describe the heterogeneity, when to expect it and when not to. Is it primarily cell-to-cell differences? Or, more on the scale of tumor-to-tumor differences?

- Figure 2: Again, it is unclear what the increase in cell-cell contacts and anchorage independence have to do with heterogeneity. These phenotypes seem consistent with increased Ras activity that would be expected in the absence of Nf2. Are these phenomena surprising or unknown prior to this?

- Figure 3: The data could be improved to better demonstrate an increase in macropinocytosis. As it is, I would say that it demonstrates an increase in dextran uptake, which *may* indicate an increase in macropinocytosis. I would expect macropinosomes to be rounder. Please note that dextran can also be taken up by receptor-mediated mechanisms.

- o Please include a brightfield (or phase) images (stills) along with the fluorescence images to better show actual macropinosomes. The supplemental video sort of helps, but the image quality is very bad. Perhaps seek some help from a microscopy expert to get better images. Also, for the figures where you are trying to demonstrate macrophages, please crop the pictures to increase the cell size so that the reader is able to see the intracellular morphologies that indicate these fluorescent spots are actually macropinosomes.

- o Also, EIPA (or amiloride) may be a problematic drug. A combination of Jasplakinolide and Blebbistatin may be a better drug combination to inhibit macropinocytosis. (Lou et al, Journal of Cell Science (2014) 127, 5228–5239 doi:10.1242/jcs.154393)

- o To show that dextran is not being taken up by a receptor-mediated mechanism, show that its uptake is not out-competed by mannan.

- o Do you see macropinocytosis heterogeneity amongst Nf2^{-/-} Schwann cells? What does MP have to do with heterogeneity?

- Figure 4.

- o The rationale for this slate of experiments could be expanded.

- o Be clear in the text of the paper whether you are talking about transcripts or protein.

- o Explain what the potential for autocrine signaling has to do with heterogeneity.

- Figure 6. This figure is finally trying to show the heterogeneity, but I don't find the data as presented sufficient to convince the reader that it is quantifiably heterogeneous. Also, please include a WT situation for comparison.

- Figure 7: In lines 252-255 you state, “Nf2^{-/-} SCs can adopt two distinct programs of coordinated ligand production and cytoskeletal polarity: High Nrg1 expression and apicojunctional polarity dominated by cortical actin ruffling and macropinocytosis, or high TGF α expression and a basal polarity program dominated by basal actin stress fibers and ECM receptors. I see a lot of individual measures of each of these things, but your case would be strengthened with microscopy showing

co-staining of both Nrg1, actin and macropinosomes in the same cell, in the two different phenotypes. And, co-staining of TGF, actin and ECM, in the two different phenotypes.

In conclusion, I find that the focus on heterogeneity is distracting and confusing. It seems that the more straight-forward goal of understanding the molecular mechanisms of schwannoma formation was met by this paper. However, to really get at heterogeneity I would recommend a different approach such as spatial RNA seq, or at the very least some co-immunostaining of proteins characterizing the distinct phenotypes.

Reviewer #3 (Remarks to the Author):

In the present manuscript, Chiasson-MacKenzie et al., investigated how heterogeneity is originated and develops in Schwannomas, using a number of different cell model systems, i.e. Schwann cells from WT and NF2^{-/-} mice, as well as Schwannoma cell lines and tissues from NF2^{-/-} mice. In vitro, the authors observed that NF2-KO in Schwann cells determines the loss of dual polarity: cells become multipolar and acquire distinct programs of ErbB ligand production (Nrg1 vs. TGF α), which correlated with distinct actin cytoskeleton rearrangements (ruffling vs. basal microvilli) and increased vs. decreased macropinocytosis. These distinct/alternative signaling programs were observed upon clonal selection of NF2^{-/-} Schwann cells, but also in different Schwannoma cell lines, showing that NF2^{-/-} cells with unstable polarity are able to self-generate autocrine signaling heterogeneity. These data were also in part confirmed in vivo in a mouse model of Schwannoma development.

The authors employ the Schwannoma as a model system to investigate an issue of high relevance for cancer cell biology, i.e. how intratumoral heterogeneity is self-generated and developed. The findings described here are novel and interesting, they describe novel biomarkers for Schwannomas' heterogeneity and might open the way to future therapeutic options using macropinocytosis to deliver therapeutic compounds.

There are, however, a number of technical, methodological and conceptual issues, listed below, that need to be solved prior publication.

Major issues:

1) Figure 1 and related Supplemental Figure 1. Methods/legends referred to these figures are too concise and lack some information. For instance, how did authors discriminate between basal vs. cortical actin in Suppl. Fig. 1c? How cortical ruffles are defined and quantified? How microvilli are defined? As both structures are marked by F-actin and ezrin, authors should show a magnification with arrows to indicate the different structures they refer to, in order to help the inexperienced reader. Please also provide further explanation in the method section.

2) Figure 1e. The bipolar appearance of WT cells is not always evident. For instance, this is clear in Fig. 1a and 1c, but it is much less evident in Fig. 1e and 2a. Maybe this could be due to different confluency, but the difference in polarity between WT and NF2^{-/-} is not always emerging. Authors should better explain this issue.

1) Figure 2c and Supplemental Figure 1d. A control for total level of ErbB2 is missing and should be provided (WB or mRNA).

2) Figure 2f and Supplemental Figure 1d. There is an apparent discrepancy between the level of pAKT in IF (where there is a clear enrichment of pAKT at the cortex of NF2^{-/-} cells, but also it seems that total pAKT level is increased in this condition) and by WB (where pAKT level remains the same in control vs NF2^{-/-} cells). I understand that this could be due to the fact that the signal is enriched at one cell location. However, to exclude unspecific staining of the antibody in IF, a control of pAKT staining in cell not stimulated with Nrg1 should be provided.

3) Figure 2h. It is very clear that the two types of cells have a different morphology and polarity. Concerning the contact with other cells, the video is not totally convincing because cells appear to be at different confluency (NF2^{-/-} cells seem at a higher confluency).

4) In Figure 4 and in general along the manuscript, authors are not systematic and perform different type of treatments in parallel experiments without a clear explanation; thus, sometimes results are not directly comparable. For instance:

i) in Figure 4a and 4e, they compared cells grown under steady-state conditions with cells deprived of Nrg1 for 24h. However, they did not perform acute stimulation of cells with Nrg1 or

TGFa or EGF as they instead did in the other panels (b, c, d). This would be helpful to have a full overview of what is going on in the different conditions.

ii) Similarly, EGF stimulation is performed in panel b and then not anymore performed along the manuscript.

iii) In panels c and d, the basal control (- Nrg1, -TGFa) is missing (at variance with panel b)

5) Figure 5. mRNA levels of Egfr, Erbb2 and Erbb3 upon glutamin depletion need to be checked. A control for the effective inhibition of ErbB activity by ErbBi needs to be provided.

6) Figure 6d-e. It would be important to show some pictures of the double labeling pS6/Nrg and pS6/TGFa, and to explain how analysis in Fig. 6e was done. Then, same analysis should be performed also for pS6/TGFa and TGFa/Nrg1 double labeling, whether possible, to see if, in that case, no correlation is observed.

7) Figure 6f. The effect of mTORi in complete medium is missing and should be provided.

8) Figure 6g. Here the authors are again not systematic: they showed the effect of ErbBi on both TGFa and Nrg1, but they showed the effect of mTORi only on Nrg1. Levels of TGFa mRNA upon mTORi should be also shown.

9) Figure 8a. It is not clear why the authors showed an overview of the different DRGs stained only for TGFa and not for Nrg1. In addition, they showed cervical vs. thoracic vs. lumbar DRGs but they do not discuss whether there is a difference among them.

In addition, I would suggest to put the pictures at low magnifications of the different types of DRGs in the supplementary (with both staining, TGFa and NRG1) and here in the main figure show some exemplary magnifications. Is it unfeasible to perform the Nrg1/TGFa double staining on these tissues? Please explain this issue.

10) Figure 8b. For a non-expert reader: it would be important to explain more details on how the DRG soma is identified and how it is defined if cells are "close to" or "far away" from the DRG soma.

11) Figure 8c. The proximity analysis should be performed also for the TGFa signal.

12) Figure 8d-e. On the same line, authors should provide the same analysis as the one shown in panel c also for the 12-months DRG lesions and show the mean cluster size (panel e) also for Nrg1, and not only for TGFa.

Minor issues

1) MW markers are missing in all WB and should be provided.

2) In Supplemental Fig. 1a p75 seems to run at a different MW in WT "SC A" as compared to the other samples. It could be a problem of running of the gel. Please explain or provide another WB.

3) Authors use cells at different confluency, e.g., "early confluent" in Fig. 1c "late confluent" in Fig. 1d. It is not immediately clear why (please explain) and what does "early and late" exactly mean.

4) In the Legend to Supplemental Fig. 2c please indicate that Cdh2 stands for the Ncad gene.

5) Data in Supplemental 3c and 3e on Schwannomas cells reconstituted with NF2 expression are striking and should be moved in main figure 3 to be more directly compared to panel 3b and panel 3e.

6) Figure 4a. It would help the reader and the flow of the manuscript to put here the mRNA level of Nrg1 in parallel to Tgfa and Egf. Then, these data can be shown again in Fig. 5b where also glutamine depletion condition is added (stating in the figure legend that they are the same).

7) The WB in Fig. 4b is not very convincing. I would not make major conclusions on this.

8) When explaining radius use μm instead of μM , as it was written in some places in the manuscript.

9) Figure 7g. Parental Nf2^{-/-} cells upon Nrg1 deprivation for 24 in comparison to a clone after long term Nrg1 deprivation are shown. Please add also the Parental Nf2^{-/-} cells in complete medium.

Chiasson-MacKenzie et al.

Point-by-point response:

We thank the reviewers for their time and thoughtful comments. In response to their suggestions and as detailed below, we have added a bolus of new experimental data, including a novel 3D model and quantitative analyses of mouse and human schwannoma tissue. We also dedicated considerable effort to better communicating the goals of our work and presenting the results in a streamlined and concise way through writing and figure management. We particularly focused on better articulating that our overarching goal was to understand how heterogeneity is self-generated within tumors using genetically homogeneous schwannomas, rather than dissecting the molecular mechanism that drives any individual phenotypic state. As Reviewer #3 notes, this is a crucial issue not just for NF2 but also for the broader cancer field. This goal is driven by an appreciation that as a cytoskeletal-associated protein, the NF2 tumor suppressor merlin does not regulate individual signaling events. Our study instead addresses the variable and context-dependent consequences of NF2-deficiency that have been reported in in vitro studies and have fractionalized therapeutic efforts. Overall, we believe that the changes guided by our reviewers have made the manuscript much stronger and of greater interest to a larger audience.

REVIEWER COMMENTS

Reviewer #1 (Remarks to the Author):

This report by two established leaders in the NF2 field focuses on understanding how heterogeneity develops in Nf2-mutant schwannomas using mouse modeling. The experiments are nicely designed, adequately powered, and appropriately interpreted.

Intrinsic polarity. Growing Schwann cells on laminin does not remove contextual cues, since laminin may provide instructive signals for polarity. This conclusion that this phenomenon is cell intrinsic should be independently validated in other contexts in vitro or using other methods.

We appreciate the positive feedback from this reviewer and entirely agree with the limitations of plating Schwann cells in standard two-dimensional conditions on laminin. To completely remove such spatial cues we developed a 3D model in which Schwann cells were embedded as single cells and allowed to expand to form large solid spheres in inert polyethylene glycol (PEG) hydrogels. As shown in a new Figure 6, in this model Nf2^{-/-} Schwann cell-derived spheroids exhibit self-generated heterogeneity in the absence of Nrg1 that is dramatically altered when external polarizing cues are provided by functionalizing the PEG gels with integrin-binding RGD peptides.

Neuregulin-1. The choice of Nrg1 was not made clear. Work from other laboratories have shown that other neuregulin family members may also be potent regulators of Schwann cell growth and motility (reviewed in Stonecypher JNEN 2006). Is it possible that other Nrg family members account for the observed heterogeneity? Similarly, prior work from the Herrlich group revealed that merlin controls CD44 ectodomain cleavage (Mol Cancer Res 2015). Was this examined, since active (cleaved NRG1) is not regulated by merlin and is predominantly involved in cell differentiation? Lastly, Fernandez-Valle showed that NRG stimulate merlin phosphorylation (Oncogene 2008).

We chose to examine Nrg1 for many reasons, including its upregulation during various other disease states (ie Fledrich et al, Nat Comm, 2019). Among Nrg1 splice isoforms we found that Nrg1 type 1 is specifically upregulated by Nf2^{-/-} Schwann cells under certain conditions (now Supplementary Figure 5e), but agree that the question of whether other Nrg family members are also heterogeneously upregulated by Nf2^{-/-} Schwann cells under certain conditions is also important. As now shown in Supplementary Figure 5e, Nrg2 is not elevated in Nf2^{-/-} SCs under any conditions. Nrg3 was not detectable by qPCR in Nf2^{-/-}

Schwann cells. We did not examine Nrg1 (or CD44) cleavage since our studies suggested that it is transcriptionally upregulated and active in conditioned medium from glutamine-deprived Nf2^{-/-} Schwann cells (Fig. 4). The reviewer makes a good point that auto/paracrine Nrg1-I produced by Nf2^{-/-} Schwann cells could influence neighboring merlin-expressing cells, ie by stimulating merlin phosphorylation, and further contribute to heterogeneity, a point which we have now added (Thaxton et al, Oncogene 2008) to the discussion.

ErbB2. How do the authors reconcile differences in their findings and those reported by Morrison and colleagues (Brain 2014)?

We don't believe there are fundamental differences between our results and those reported by Morrison and colleagues (Schulz et al 2014), which we had mentioned in the Discussion, as the two studies focused on very different things using different models. Schulz et al examined neuronal Nf2-deficiency or constitutional isoform-specific Nf2-deficiency, while we focused exclusively on Schwann cell-specific Nf2-deficiency to study intrinsic aspects of tumor initiation and heterogeneity. Schulz et al conclude that neuronal Nf2-deficiency yields decreased neuronal Nrg1 and increased ErbB2 levels on associated myelinating wild-type Schwann cells in vivo. Our study focuses on the ability of Nf2-deficient Schwann cells to over-react to exogenously provided Nrg1 or to over-produce their own Nrg1 under conditions designed to mimic the loss of neuronal contact in vitro and in tumor tissue. Like Schulz et al, we see a modest upregulation of ErbB2 mRNA upon Nrg deprivation (but not glutamine deprivation as shown in FigS5D), but in BOTH WT and Nf2^{-/-} SCs (not shown). The obvious conclusion from both studies is that the neuroglial interface involves complex reciprocal signaling that may be influenced by the genetics of both neuron and Schwann cell, which is exactly why we turned to in vivo multiplex analysis of tumor tissue to study it.

Mechanistic insights. It is necessary to demonstrate cause and effect relationships between merlin and ErbB function/signaling. This is particularly relevant to dissecting the mechanism underlying merlin regulation of pinocytosis.

The goal of our study wasn't at all to define the mechanism by which loss of NF2 influences ErbB2/3 function/signaling or macropinocytosis when glutamine deprived or EGFR signaling when Nrg-deprived; these are complex questions that were subjects of earlier work (Curto et al 2007, Chiasson-MacKenzie et al 2015, 2018 and others). Indeed, it is clear that merlin is an unusual tumor suppressor in being a cytoskeletal-associated protein that does not function in a single linear signaling pathway, and the consequences of its loss are context- and cell type-dependent. With this in mind, the focus of the current study was to understand how heterogeneity develops in genetically cold Nf2^{-/-} schwannomas. As this focus was also not clear to Reviewer #2, we have reworded and streamlined the results to clarify this, moving the ErbB2-specific data to the Supplement.

Autocrine heterogeneity. In this section, the authors present a series of observations, but do not provide conclusive etiologic evidence for any one of the mechanisms as being responsible for the phenotypes seen in Nf2-deficient Schwann cells.

Again, our goal was to understand how heterogeneity develops intrinsically within schwannomas in the presence of a single genetic driver, as opposed to the mechanistic basis of how the overreactive cytoskeletal response causes the overproduction of one or the other ligand, which would be beyond the scope of the current manuscript. We have now mentioned this as a clear avenue of follow up in the Discussion.

mTOR dependence. A recent study also demonstrated that neuregulin-ErbB3 autocrine signaling in NF2-deficient tumor cells is controlled by mTOR (J Biol Chem 2021). It is not clear why the authors conclude that pS6 is a specific biomarker of autocrine Nrg1 signaling.

We believe that Figure 5 (previously Figure 6) provides strong evidence that pS6 is a biomarker of auto/paracrine Nrg1 production in Schwann cells: Nrg1 alone and conditioned medium from cells expressing high levels of Nrg1 induces pS6. In addition, we now provide evidence in new Figures 6, 7 and 8, that Nrg1 induces pS6 in a 3D model, and that Nrg1 and pS6 codistribute in human and mouse schwannoma tumor tissue, also supporting this conclusion. The important distinction between our studies and those of meningioma, which we had referenced in the Discussion (Beauchamp et al JBC, 2021), is that mTOR inhibition does not block the dramatically elevated Nrg1 expression triggered by glutamine deprivation in cultured Schwann cells. Important follow-up studies will deploy quantitative imaging to examine Nrg1 and pS6 levels and distribution in mTORi-treated mice and, ultimately humans.

Heterogeneity in tumors. It would be helpful to demonstrate that Nrg1+ and TGF α + cells have a similar pattern in human tumors. In addition, since both proteins are secreted, how confident are the investigators that they are capturing cell expression? RNAScope would be a more definitive method to demonstrate which cells are making these paracrine factors. It is also not clear in Figure 7 which cells in the tumor are expressing these factors.

We have examined human tumors and, as is now shown in Figure 7, found that Nrg1 and pS6 codistribute, as predicted by our in vitro studies and also seen in mouse schwannoma tissue (Fig. 8), and that the distribution of Nrg+/pS6+ cells anti-correlates with that of cells exhibiting high levels of the 'basal' marker pNDRG in human and mouse schwannoma tissue (Fig 8). Unfortunately, we were unable to find antibodies that detect human TGF α for immunofluorescence despite testing several. We also developed RNAScope as a technique in the lab and validated the distribution of Nrg1 and TGF α protein and mRNA within mouse schwannoma tissue as is now shown in Figure 8 and Supplemental Figure 9.

Reviewer #2 (Remarks to the Author):

The goal of this study was to characterize the heterogeneity seen in Nf2 $^{-/-}$ Schwann cells and determine the underlying molecular mechanisms of this heterogeneity. The authors characterized the phenotype of Nf2 $^{-/-}$ Schwann cells and found them to have increased multipolarity, increased actin polymerization, increased cell-cell contacts, anchorage independence, increased dextran uptake (proxy for macropinocytosis), and increased autocrine signaling. They suggested that the autocrine signaling could set up some kind of stochastic process whereby cells become polarized to distinct phenotypes. However, I found that the focus on heterogeneity is distracting and confusing. The relationship between heterogeneity, multipolarity, increased actin polymerization, increased cell-cell contacts, etc. was not articulated or demonstrated. I feel it would be more straight-forward for the authors to state the goal of the study was to simply understand the molecular mechanisms of schwannoma pathology.

We thank the reviewer for his/her comments, which prompted us to focus intensely on clarifying our goal and presenting the results in a more streamlined and concise manner. Our overarching goal was and is to understand how tumor heterogeneity develops in the presence of a single genetic event, which is a striking feature of schwannoma pathology and clinical behavior that impedes translational efforts for these patients. Appreciation of the importance of intrinsic tumor heterogeneity is emerging from studies of other cancers but must be inferred from large -omics studies of tumors with complex genomes and microenvironments. The simple genetics of schwannoma allow insight into this also inherently complex biology. As the NF2 tumor suppressor is a cytoskeletal protein, it does not govern a single 'pathway' and the consequences of its loss are context-dependent, as corroborated by our in vitro studies. We argue that the propensity for intrinsic heterogeneity is itself central to the molecular mechanism of schwannoma pathology. Although dissection of the stochastic (or microenvironmentally biased) process by which Nf2 $^{-/-}$ cells 'become polarized to distinct phenotypes' is interesting and important mechanistic follow-up, that

was not our goal here. Rather than using an unbiased approach to identify biomarkers of schwannoma heterogeneity that would require functional follow-up, we deployed fundamental cell biology approaches to identify distinct 'states' that Nf2^{-/-} SCs adopt depending on the availability of physiological nutrients. We then validated biomarkers of these states in human and mouse schwannoma tumors. We spent considerable time describing the phenotypes of these two 'states' as the information may be relevant translationally, but agree that it was distracting, so we significantly condensed these descriptions and believe that this, together with the important new demonstration of intrinsic heterogeneity in 3D and validation in human and mouse schwannoma tissue, yield a much more focused message.

- I'm not sure that I accept the claim that "An understanding of the molecular basis of schwannoma heterogeneity is essential for the development of successful non-surgical therapies." (Lines 51-53). If this were the case, then you would need to provide evidence from previous work showing that some phenotypes are treatable while others are not. Rather, is it that a lack of successful therapies is simply due to insufficient understanding of the molecular mechanisms of disease?

It is well known that the clinical behavior and therapeutic response of schwannomas is heterogeneous, even when the tumors arise within the same NF2 patient and without cooperating genetic events. In fact, most NF2 patients develop schwannomas on each vestibular nerve and often the clinical behavior and therapeutic response of the two tumors is different (for example Plotkin et al 2008). In addition to this inter-tumoral heterogeneity, the few drugs that have been evaluated in schwannoma patients yield at best a cytostatic response in some tumors despite the fact that all schwannoma cells in a tissue culture dish under a specific condition, are sensitive. On top of that we have now demonstrated significant intra-tumoral heterogeneity in schwannoma tissue, including for biomarkers (pS6 and pNDRG) of drugs that are actively being clinically evaluated. All of this supports our contention that understanding this heterogeneity and studying it in vivo is essential for the development of successful non-surgical therapies. In fact, 'Mechanisms of Heterogeneity' has been listed as a specific 'Area of Emphasis' in the DOD Neurofibromatosis Research Program RFA for several years. We have updated our language to better emphasize this.

- General methodology questions:

- o Why use RNAi when you could use a CRISPR gene disruption system?

There are obvious pros and cons to using either RNAi or CRISPR. The only experiment in the paper involving either is the shRNA-based knockdown of ezrin, for which we had validated reagents in hand (Chiasson-MacKenzie et al, 2018).

- o I do not have expertise in Schwann cell culture and cannot comment on the experimental model system of isolating cells from spontaneously generated mouse tumors in KO animals. However, as study of cellular heterogeneity seems especially sensitive to cellular context, differentiation, and so forth, it would be helpful if the authors provided justification or validation for their model system. In the area of cell biology that I work in, we would expect dramatically different phenotypes in cells isolated from different areas of the body, from mice of different ages and so forth. Can you more clearly state how this work is novel in that regard and how your experimental set-up is justified to study this question?

The majority of our in vitro studies were carried out in non-tumor-derived mouse Schwann cells isolated and cultured under standard conditions and in which Nf2 was acutely deleted in culture via Cre-lox technology. The heterogeneous phenotypes are driven specifically by changing nutrient availability as opposed to the origin of the cells, and are not exhibited by parental (wild-type) Schwann cells. We have now provided additional evidence that Nf2^{-/-} Schwann cells can self-generate heterogeneity in a new 3D model, further supporting the idea that NF2-deficiency itself enables heterogeneity. This has not been described before.

- Figure 1: This figure demonstrates the increased multipolarity, change in cellular morphology and increased cortical actin in the *Nf2*^{-/-} cells, but it doesn't show exceptional heterogeneity. It would help to describe the heterogeneity, when to expect it and when not to. Is it primarily cell-to-cell differences? Or, more on the scale of tumor-to-tumor differences?

*As described above, these are non-tumor murine Schwann cells in which *Nf2* has or has not been acutely deleted in vitro. The heterogeneity is not cell to cell but condition to condition in vitro. Figure 1 shows that the acutely deleted *Nf2*^{-/-} cells but not the WT cells uniformly exhibit multipolarity, which is easiest to see when they are plated sparsely. *Nf2*^{-/-} Schwann cells adopt one of two completely different cytoskeletal phenotypes depending on the availability of *Nrg1*: in the presence of *Nrg1* *Nf2*^{-/-} cells exhibit strong cortical actin and ezrin-containing ruffles and in the absence of *Nrg1* *Nf2*^{-/-} SCs instead form a basally oriented cytoskeleton dominated by actin stress fibers and ezrin-containing microvilli. The wild-type cells show little change in phenotype under these conditions. Later in the paper, we linked these phenotypes to auto/paracrine ligand production, which suggests that they may enable self-generated intra-tumoral heterogeneity in vivo, which we then validated in tumor tissue and a 3D model. We hypothesize that variations in the extent of intra-tumoral heterogeneity in turn, leads eventually to inter-tumoral (tumor-to-tumor) heterogeneity, which we began to see in our mouse model even at early timepoints. In streamlining the results we have tried to better explain this and forestall confusion.*

- Figure 2: Again, it is unclear what the increase in cell-cell contacts and anchorage independence have to do with heterogeneity. These phenotypes seem consistent with increased Ras activity that would be expected in the absence of *Nf2*. Are these phenomena surprising or unknown prior to this?

*The data shown in Figure 2, which we have now combined with Figure 3, describe the 'state' that *Nf2*^{-/-} Schwann cells enter in the presence of *Nrg1* only; in the absence of *Nrg1* they enter a different 'state' dominated by a basally oriented cytoskeleton and EGF ligand production. These are not tumor cells and since schwannomas are benign tumors we would expect neither to be 'transformed' like Ras-mutant cancer cells. Moreover, non-tumor Schwann cells that are mutant for the neurofibromatosis type 1 (*NF1*) tumor suppressor, which is known to be a Ras-GAP, do not show these phenotypes (Chiasson-MacKenzie et al, Genes Dev 2018). These data are particularly surprising given that we and others have shown that other types of *Nf2*^{-/-} cells exhibit defective cell-cell contact rather than increased cell-cell contact.*

- Figure 3: The data could be improved to better demonstrate an increase in macropinocytosis. As it is, I would say that it demonstrates an increase in dextran uptake, which *may* indicate an increase in macropinocytosis. I would expect macropinosomes to be rounder. Please note that dextran can also be taken up by receptor-mediated mechanisms.

*We have moved some of this data into Figure 2 and moved the rest to the Supplemental data. Our previously published work focused on increased macropinocytosis exhibited by other types of *Nf2*^{-/-} cells (Chiasson-MacKenzie et al, Genes Dev 2018). Here we were only showing it as an additional readout of the *Nrg1*⁺ but not the *Nrg1*⁻ state. Nevertheless, we have added brightfield images and increased the cell sizes as requested.*

o Please include a brightfield (or phase) images (stills) along with the fluorescence images to better show actual macropinosomes. The supplemental video sort of helps, but the image quality is very bad. Perhaps seek some help from a microscopy expert to get better images. Also, for the figures where you are trying to demonstrate macrophages, please crop the pictures to increase the cell size so that the reader is able to see the intracellular morphologies that indicate these fluorescent spots are actually macropinosomes.

*Again, increased macropinocytosis was only meant to be an additional, potentially translational feature of the *Nrg1*⁺ 'state' that we had already linked to *Nf2*-deficiency in other cell types in a previous publication.*

To allay this reviewers' concern over specificity we have referred to macropinocytosis as 'EIPA-sensitive dextran-488 uptake' or 'a macropinocytosis-like process' in the now-condensed results. We have also replaced the supplemental video with a brightfield confocal image overlaid with dextran-488 to show that dextran-488 localizes to macropinosome-like structures upon Nrg1 stimulation.

o Do you see macropinocytosis heterogeneity amongst Nf2^{-/-} Schwann cells? What does MP have to do with heterogeneity?

As described above, the heterogeneity we are focusing on is not cell-to-cell heterogeneity in vitro and is instead driven by changing nutrient availability. Macropinocytosis is an additional readout of the Nrg1⁺ state but not the Nrg1⁻ (basal) state in vitro and therefore, like pS6, a potential biomarker of Nrg1⁺ regions of the tumor in vivo. The fact that macropinocytosis may enable macrotherapeutic uptake makes it a therapeutically relevant biomarker. In regions of the tumor that are Nrg1⁺/pS6⁺/macropinocytic may preferentially take up larger drugs relative to regions of the tumor that are marked by basal state markers (TGF α , pNDRG1, Itga6 etc).

• Figure 4.

o The rationale for this slate of experiments could be expanded.

We have more clearly stated the rationale for these experiments, which demonstrate that in the absence of Nrg1 the Nf2^{-/-} Schwann cells upregulate EGF ligands rather than Nrg1, and that EGF ligands drive the basal cytoskeletal phenotype.

o Be clear in the text of the paper whether you are talking about transcripts or protein.

We have scanned the paper to make sure that we always adhered to the use of italics for mRNA but not protein, and lower case for mouse/upper case for human, and looked for places where we should even more explicitly clarify this point.

o Explain what the potential for autocrine signaling has to do with heterogeneity.

We have clarified that the ability of Nf2^{-/-} Schwann cells to, depending on their context, produce different growth factors that drive distinct phenotypes in vitro, may enable 'self-generated' heterogeneity in the form of neighborhoods within a tumor composed of cells of a certain state. Our validation in a 3D model, as well as in human and particularly mouse tumors highlights the complexity of the resulting patterns of heterogeneity and the need to understand how they develop and progress.

• Figure 6. This figure is finally trying to show the heterogeneity, but I don't find the data as presented sufficient to convince the reader that it is quantifiably heterogeneous. Also, please include a WT situation for comparison.

We introduced a novel 3D model of self-generated heterogeneity as a new Figure 6. In addition, we removed panels d and e from this figure (previously Figure 6) and instead introduced quantitative imaging of human and mouse schwannoma tissue as new Figures 7 and 8, respectively. In the large human tumors we show that biomarkers of the two phenotypic states that we identified in vitro exhibit regionally and quantifiably heterogeneous distribution in human schwannomas, with codistribution of Nrg and pS6 and anti-correlation between Nrg/pS6 and pNDRG as predicted by our in vitro studies. Analysis of developing mouse tumors yields similar results and novel insight into how intrinsic intratumor patterns of heterogeneity begin.

• Figure 7: In lines 252-255 you state, "Nf2^{-/-} SCs can adopt two distinct programs of coordinated ligand

production and cytoskeletal polarity: High Nrg1 expression and apicojunctional polarity dominated by cortical actin ruffling and macropinocytosis, or high TGF α expression and a basal polarity program dominated by basal actin stress fibers and ECM receptors. I see a lot of individual measures of each of these things, but your case would be strengthened with microscopy showing co-staining of both Nrg1, actin and macropinosomes in the same cell, in the two different phenotypes. And, co-staining of TGF, actin and ECM, in the two different phenotypes.

In now Figure 8, we provide a completely new and quantitative analysis of multiple biomarkers of heterogeneity predicted by our in vitro studies in developing schwannomas in the well-established mouse model. As in human tumors, this analysis revealed regional concordance for Nrg1 and pS6 and anticorrelation between Nrg+/pS6+ and pNDRG1+ cells. Furthermore, the mapping of our biomarkers onto well-known cellular patterns in early lesions provided an exciting glimpse into how spatial patterns of heterogeneity arise and progress in tumors, and a foundation for understanding how those patterns progress and are influenced by drug treatment.

In conclusion, I find that the focus on heterogeneity is distracting and confusing. It seems that the more straight-forward goal of understanding the molecular mechanisms of schwannoma formation was met by this paper. However, to really get at heterogeneity I would recommend a different approach such as spatial RNA seq, or at the very least some co-immunostaining of proteins characterizing the distinct phenotypes.

We appreciate this reviewers comments as they prompted us to simplify and focus our message for the benefit of a broader audience. The goal of quantitatively and spatially mapping multiple biomarkers of heterogeneity in tumors was always our top priority and has now been added.

Reviewer #3 (Remarks to the Author):

In the present manuscript, Chiasson-MacKenzie et al., investigated how heterogeneity is originated and develops in Schwannomas, using a number of different cell model systems, i.e. Schwann cells from WT and NF2^{-/-} mice, as well as Schwannoma cell lines and tissues from NF2^{-/-} mice. In vitro, the authors observed that NF2-KO in Schwann cells determines the loss of dual polarity: cells become multipolar and acquire distinct programs of ErbB ligand production (Nrg1 vs. TGF α), which correlated with distinct actin cytoskeleton rearrangements (ruffling vs. basal microvilli) and increased vs. decreased macropinocytosis. These distinct/alternative signaling programs were observed upon clonal selection of NF2^{-/-} Schwann cells, but also in different Schwannoma cell lines, showing that NF2^{-/-} cells with unstable polarity are able to self-generate autocrine signaling heterogeneity. These data were also in part confirmed in vivo in a mouse model of Schwannoma development.

The authors employ the Schwannoma as a model system to investigate an issue of high relevance for cancer cell biology, i.e. how intratumoral heterogeneity is self-generated and developed. The findings described here are novel and interesting, they describe novel biomarkers for Schwannomas' heterogeneity and might open the way to future therapeutic options using macropinocytosis to deliver therapeutic compounds.

We thank this reviewer for his/her positive comments and for articulating our overarching goal so well.

There are, however, a number of technical, methodological and conceptual issues, listed below, that need to be solved prior publication.

Major issues:

1) Figure 1 and related Supplemental Figure 1. Methods/legends referred to these figures are too concise and lack some information. For instance, how did authors discriminate between basal vs. cortical actin in

Suppl. Fig. 1c? How cortical ruffles are defined and quantified? How microvilli are defined? As both structures are marked by F-actin and ezrin, authors should show a magnification with arrows to indicate the different structures they refer to, in order to help the inexperienced reader. Please also provide further explanation in the method section.

As we have now clarified in the Methods and legends to Figure 1 and Supplementary Figure 1, Cortical actin and basal ezrin area were defined using an approach similar to that described in Condon et al, JCB 2018, which has now been added as a reference. Briefly, in z stack images stained with phalloidin to label F-actin, anti-ezrin antibodies, and DAPI to label nuclei, the nuclear midpoint was determined and the F-actin labeled area above was defined as cortical actin and the ezrin labeled region below was defined as basal ezrin area. Cortical ruffles were defined as actin-containing structures that extended at least 5 μm above the midpoint of the nucleus in z stack images. The images in Supplementary Fig. 1B were created by overlaying a maximum intensity projection of the cell volume above the nuclear midpoint (cortical actin) with a maximum intensity projection of the cell volume below the nuclear midpoint (basal actin) from 3D z stack images. Microvilli were defined as ezrin and F-actin labeled structures localized to boundaries between neighboring cells as the basal surface of the cell (below the nuclear midpoint). A higher magnification inset depicting actin ruffles and microvilli is now provided in Supplementary Fig. 1C.

2) Figure 1e. The bipolar appearance of WT cells is not always evident. For instance, this is clear in Fig. 1a and 1c, but it is much less evident in Fig. 1e and 2a. Maybe this could be due to different confluency, but the difference in polarity between WT and NF2^{-/-} is not always emerging. Authors should better explain this issue.

Indeed, it is much harder to see the bipolarity of wild-type cells at confluence for multiple reasons. To aid visualization we have outlined individual cells in Fig. 1 D and F to better show that both cell shape (aspect ratio) and surface structures (ruffles or microvilli) are bipolar in wild-type and radial in Nf2^{-/-} Schwann cells.

1) Figure 2c and Supplemental Figure 1d. A control for total level of ErbB2 is missing and should be provided (WB or mRNA).

We have added data showing similar levels of ErbB2 and ErbB3 mRNA in wild-type and Nf2^{-/-} SCs to Supplementary Figure 2D.

2) Figure 2f and Supplemental Figure 1d. There is an apparent discrepancy between the level of pAKT in IF (where there is a clear enrichment of pAKT at the cortex of Nf2^{-/-} cells, but also it seems that total pAKT level is increased in this condition) and by WB (where pAKT level remains the same in control vs NF2^{-/-} cells). I understand that this could be due to the fact that the signal is enriched at one cell location. However, to exclude unspecific staining of the antibody in IF, a control of pAKT staining in cell not stimulated with Nrg1 should be provided.

Although we moved the detailed analysis of ErbB/pAKT into Supplementary Figure 2 to maintain emphasis on heterogeneity rather than the mechanistic basis of any one phenotypic state, we do think that differences in the spatial distribution of membrane receptor signaling is a mechanistic hallmark of Nf2^{-/-} deficiency. We have added images showing pAkt in unstimulated wild-type and Nf2^{-/-} SCs to Supplementary Fig. 2E.

3) Figure 2h. It is very clear that the two types of cells have a different morphology and polarity. Concerning the contact with other cells, the video is not totally convincing because cells appear to be at different confluency (NF2^{-/-} cells seem at a higher confluency).

This is because the control cells, but not the Nf2^{-/-} SCs bounce off of each other and accelerate in a different direction instead of sticking to one another, therefore maintaining greater intercellular distances,

which is the classic presentation of contact inhibition of locomotion (e.g. Carmona-Fontaine et al 2008). This is why we showed the time-lapse images from a movie and then quantified the time of contact over several movies (now in Figure 2D,E). We have clarified this in the text.

4) In Figure 4 and in general along the manuscript, authors are not systematic and perform different type of treatments in parallel experiments without a clear explanation; thus, sometimes results are not directly comparable. For instance:

i) in Figure 4a and 4e, they compared cells grown under steady-state conditions with cells deprived of Nrg1 for 24h. However, they did not perform acute stimulation of cells with Nrg1 or TGF α or EGF as they instead did in the other panels (b, c, d). This would be helpful to have a full overview of what is going on in the different conditions.

We would not expect changes in mRNA expression to occur in the same short time period (<30 minutes) in which redistribution of actin and macropinocytosis occurs in response to acute stimulation. Therefore, we allowed 24 hours of changed nutrient environment (ie Nrg or glutamine deprivation) to see elevation of Nrg1 or TGF α mRNA. Note that steady state conditions include Nrg1 in the medium.

ii) Similarly, EGF stimulation is performed in panel b and then not anymore performed along the manuscript.

Since TGF α was the dominant EGF ligand upregulated by Nf2^{-/-} SCs upon Nrg deprivation and the more highly expressed of the two ligands, and EGF and TGF α induced similar changes in macropinocytosis and visible ruffling, we felt that they were largely redundant.

iii) In panels c and d, the basal control (- Nrg1, -TGF α) is missing (at variance with panel b)

In order to simplify the figure, and since the controls are the same as in Figure 1E,F, we didn't include the images again in Figure 3 (formerly Figure 4) but have now referred back to them in the text.

5) Figure 5. mRNA levels of Egfr, ErbB2 and ErbB3 upon glutamin depletion need to be checked. A control for the effective inhibition of ErbB activity by ErbBi needs to be provided.

We have now added qPCR data showing that glutamine deprivation does not alter the expression of EGFR, ErbB2 or ErbB3 to Supplementary Fig. 5D.

6) Figure 6d-e. It would be important to show some pictures of the double labeling pS6/Nrg and pS6/TGF α , and to explain how analysis in Fig. 6e was done. Then, same analysis should be performed also for pS6/TGF α and TGF α /Nrg1 double labeling, whether possible, to see if, in that case, no correlation is observed.

We have focused much attention on this aspect of the manuscript. We removed panels 6d-e and replaced them with a more extensive analysis of human (Figure 7) and mouse (Figure 8) schwannoma tissue. We are currently limited in our ability to stain for multiple markers in a single tissue section due to lack of antibody compatibility and an inability to obtain a human anti-TGF α antibody suitable for IF. We are currently developing tyramide signal amplification (TSA)-based multiplexed imaging using Opal dyes so that we can examine all of these biomarkers simultaneously, but this will take significant time and resources that are beyond the scope of the current manuscript. However, we used multiple approaches to overcome these technical hurdles. First, we carried out neighborhood based spatial correlation analysis of NRG1, pS6 and pNDRG1 (which is induced basally by integrin attachment in Schwann cells in vivo (Heller et al, 2014)) in human and mouse tissue using serial tissue sections. The analyses in Figures 7 and 8 show within human and mouse schwannomas, that the distribution of Nrg1 and pS6 significantly correlate and that both Nrg1 and pS6 anticorrelate with pNDRG, as predicted by our in vitro studies.

Second, we developed FISH and FISH-IF using RNAscope technology to examine Nrg1/Nrg1 and TGF α /Tgfa mRNA and protein distribution in mouse DRG tissues. As shown in Figure 8, we found that when analyzed simultaneously, Nrg1 and Tgfa mRNAs were anticorrelated and that Nrg1 mRNA, like protein, exhibited spatial proximity to DRG soma.

7) Figure 6f. The effect of mTORi in complete medium is missing and should be provided.

We have now added data to Supplementary Fig. 6 showing that cells in complete medium are relatively resistant to mTORi.

8) Figure 6g. Here the authors are again not systematic: they showed the effect of ErbBi on both TGF α and Nrg1, but they showed the effect of mTORi only on Nrg1. Levels of TGF α mRNA upon mTORi should be also shown.

We have now added data as Fig. 5e showing that everolimus modestly blocks Tgfa upregulation under conditions of Nrg1 deprivation.

9) Figure 8a. It is not clear why the authors showed an overview of the different DRGs stained only for TGF α and not for Nrg1. In addition, they showed cervical vs. thoracic vs. lumbar DRGs but they do not discuss whether there is a difference among them.

We showed the overview just to illustrate the power of the model in our ability to quantitatively analyze multiple DRG per mouse simultaneously. Although we did not see major differences among DRG from different anatomical locations at these early timepoints a larger cohort of mice of the same age would be required to statistically address that question.

In addition, I would suggest to put the pictures at low magnifications of the different types of DRGs in the supplementary (with both staining, TGF α and NRG1) and here in the main figure show some exemplary magnifications. Is it unfeasible to perform the Nrg1/TGF α double staining on these tissues? Please explain this issue.

Indeed, Nrg1/TGF α double staining is complicated as the most specific antibodies are both polyclonal rabbit antibodies and we have been unable to identify an anti-TGF α antibody that works well on human tissue. Instead, we monitored pNDRG1, which is known to be induced in SCs by integrin α 6, a biomarker of the basal phenotype; indeed, we found pNDRG1 to be anti-correlated with Nrg1+/pS6+ cells in both human and mouse tumor tissue and with pS6 in a new 3D model that we developed and added to the revised manuscript. In addition, we developed FISH and FISH-IF using RNAscope and used it to examine the distributions of Nrg1 and Tgfa mRNAs in mouse DRG lesions. As shown in Figure 8 and Supplementary Figure 9, we found that Nrg1 and Tgfa mRNAs were anticorrelated and heterogeneous across all DRGs and that Nrg1 mRNA, like protein, exhibited spatial proximity to DRG soma.

10) Figure 8b. For a non-expert reader: it would be important to explain more details on how the DRG soma is identified and how it is defined if cells are “close to” or “far away” from the DRG soma.

The DRG soma are easily identified morphologically and we trained the HALO software to recognize them and exclude them for the quantitative analysis or use them as landmarks for proximity analysis. In the latter case, cells within 25 μ m of individual soma were quantified. We have more explicitly stated this in the methods.

11) Figure 8c. The proximity analysis should be performed also for the TGF α signal.

As part of the FISH/FISH-IF analysis by RNAscope we now show that Nrg1 and TGFa mRNA distribution is anti-correlated in Fig. 8 D,E and Supplementary Fig 9 C, D.

12) Figure 8d-e. On the same line, authors should provide the same analysis as the one shown in panel c also for the 12-months DRG lesions and show the mean cluster size (panel e) also for Nrg1, and not only for TGFa.

We have removed this analysis for Tgfα and replaced it with RNAscope analysis. Future quantitative analyses of the evolution of inter-tumoral heterogeneity across DRGs should include a much larger cohort of older animals.

Minor issues

1) MW markers are missing in all WB and should be provided.

We have added these.

2) In Supplemental Fig. 1a p75 seems to run at a different MW in WT “SC A” as compared to the other samples. It could be a problem of running of the gel. Please explain or provide another WB.

Indeed, this is a gel running problem; we have provided another immunoblot.

3) Authors use cells at different confluency, e.g., “early confluent” in Fig. 1c “late confluent” in Fig. 1d. It is not immediately clear why (please explain) and what does “early and late” exactly mean.

We have added definitions of ‘early confluence’ and ‘late confluence’ to the methods.

4) In the Legend to Supplemental Fig. 2c please indicate that Cdh2 stands for the Ncad gene.

We have updated the Supplementary Fig 2 legend (now panel g) to clarify this.

5) Data in Supplemental 3c and 3e on Schwannomas cells reconstituted with NF2 expression are striking and should be moved in main figure 3 to be more directly compared to panel 3b and panel 3e.

Although we agree that this is a striking result, we feel that it would be difficult to move it to the main figures given our efforts to streamline the manuscript and message in response to the reviewers’ suggestions. We have combined Figures 2 and 3, added new data and moved quite a bit of additional data to the Supplementary Figures.

6) Figure 4a. It would help the reader and the flow of the manuscript to put here the mRNA level of Nrg1 in parallel to Tgfα and Egf. Then, these data can be shown again in Fig. 5b where also glutamine depletion condition is added (stating in the figure legend that they are the same).

We have now added mRNA levels of Nrg1 under conditions of Nrg1 withdrawal as Fig. 3A

7) The WB in Fig. 4b is not very convincing. I would not make major conclusions on this.

We have removed this data and replaced it with qPCR data showing that Col1a2 is also upregulated.

8) When explaining radius use μm instead of μM, as it was written in some places in the manuscript.

Thank you for catching this. We have corrected it.

9) Figure 7g. Parental Nf2^{-/-} cells upon Nrg1 deprivation for 24 in comparison to a clone after long term Nrg1 deprivation are shown. Please add also the Parental Nf2^{-/-} cells in complete medium.

To make room for the new 3D model and human and mouse tissue analyses we have removed the studies of our clonal mouse schwannoma cell lines from the manuscript.

REVIEWER COMMENTS

Reviewer #1 (Remarks to the Author):

There were several issues raised in the original review.

Intrinsic polarity. The authors responded well to the original comments by developing a 3D model using PEG hydrogels.

Neuregulin-1. While the new data on neuregulin family member expression provide a rationale for focusing on Nrg1, the comment that "its upregulation during various other disease states" is not convincing or relevant to tumorigenesis. Moreover, why does transcriptional regulation exclude the possibility that the secreted NRG1 is cleaved and acts to control merlin function? Is the Nrg1 commercially purchased a full-length protein? This should be clarified.

ErbB2. This comment was adequately addressed.

Mechanistic insights. While no mechanistic insights were provided, the authors have restricted the manuscript to more clearly articulate the goals of the study.

mTOR dependence. mTOR signaling, as the authors state, is complex. mTOR is a complex (TORC1, TORC2, TORC3) with different molecular binding partners. In this regard, is it not possible that S6 is controlled by mTOR in an everolimus-independent manner? The use of genetic silencing (raptor, S6K) or PI3K inhibitors would be more definitive, rather than relying on a single pharmacologic inhibitor. Was there no change in 4EBP1 phosphorylation to support an mTOR-independent mechanism? Did the authors examine S6K1 or RSK activity, as there are other kinases that phosphorylate S6 on Serine residues 235/236?

Heterogeneity in tumors. This comment was adequately addressed.

Reviewer #2 (Remarks to the Author):

I recommend the manuscript, "Cellular mechanisms of heterogeneity in NF2-mutant schwannoma" by Christine Chiasson-MacKenzie for publication. I appreciate the clarifications made to explaining and justifying their experimental model for studying context-dependent heterogeneity in Schwann cells. Overall, the quality of the data and clarity of the manuscript is much improved and makes significant progress toward advancing understanding how context-specific signals program Schwann cells to distinct phenotypes. It still feels like the reader needs to work hard in some places to follow the rationale and conclusions of the experiments. Some simple improvements like making the abstract more mechanistic and improving some of the figure legends (e.g. Figure 2) may help.

Reviewer #3 (Remarks to the Author):

The authors have provided a number of additional experiments and text revisions that addressed my major issues.

I think the manuscript have improved and is now suitable for publication.

I have only few minor points left, listed below.

- In Fig. 1 d, 1f please explain in the figure legend how you defined the cell border
- Fig. 3e. Authors need to include the non-treated control coming from the same experiment and not simply referring to the control present in Fig. 1e, 1g (the absolute values vary from experiment to experiment, as also visible by comparing these two experiments).
- In Fig. 8b left is not clear to me why the Nrg+ cells are not pink. Please add more explanations in the figure legend.
- When explaining radius there is still a "uM" left at line 632 that need to be changed to um.

Point-by-point
Chiasson-MacKenzie et al.

Reviewer #1 (Remarks to the Author):

There were several issues raised in the original review.

We thank this reviewer for her/his insightful and constructive comments throughout.

Intrinsic polarity. The authors responded well to the original comments by developing a 3D model using PEG hydrogels.

Neuregulin-1. While the new data on neuregulin family member expression provide a rationale for focusing on Nrg1, the comment that “its upregulation during various other disease states” is not convincing or relevant to tumorigenesis. Moreover, why does transcriptional regulation exclude the possibility that the secreted NRG1 is cleaved and acts to control merlin function? Is the Nrg1 commercially purchased a full-length protein? This should be clarified.

Our comment about Nrg1 upregulation in other disease states was not included in the manuscript but we actually do believe that onion bulb formation in peripheral neuropathies (Fledrich et al 2019) may be very relevant to schwannoma. The reviewer is correct that transcriptional upregulation of Nrg1 does not exclude the possibility of other layers of regulation and bystander effects on neighboring merlin-expressing cells, which is a good idea that we did acknowledge and now have expanded upon in the Discussion. Our studies were entirely focused on Nf2^{-/-} (merlin-deficient) cells. We have indicated that commercially available Nrg1 is a bioactive fragment in the Methods.

ErbB2. This comment was adequately addressed.

Mechanistic insights. While no mechanistic insights were provided, the authors have restricted the manuscript to more clearly articulate the goals of the study.

mTOR dependence. mTOR signaling, as the authors state, is complex. mTOR is a complex (TORC1, TORC2, TORC3) with different molecular binding partners. In this regard, is it not possible that S6 is controlled by mTOR in an everolimus-independent manner? The use of gene silencing (raptor, S6K) or PI3K inhibitors would be more definitive, rather than relying on a single pharmacologic inhibitor. Was there no change in 4EBP1 phosphorylation to support an mTOR-independent mechanism? Did the authors examine S6K1 or RSK activity, as there are other kinases that phosphorylate S6 on Serine residues 235/236?

Our point-by-point addresses this reviewers' original question of why we ‘...conclude that pS6 is a specific biomarker of autocrine Nrg1 signaling’ by presenting multiple lines of evidence that Nrg1 induces pS6 activity in our cells and tissues, in addition to referencing published evidence that axonal Nrg1 induces pS6 in normal polarized Schwann cells in vivo (Heller et al 2014). We did not focus on the different questions of whether pS6 is a specific readout for mTORC1 activation or is activated in an mTOR-dependent, everolimus-independent manner. Others have shown that pS6 is an indicator of mTORC1 activity in Schwann cells and schwannoma as it is in many cell/tissue types; in fact, in that same study of normal Schwann cells (Heller et al 2014), Nrg1-induced pS6 was shown to be PI3K-mTORC1-dependent and specifically everolimus-sensitive. Moreover, everolimus treatment reduces pS6 levels in in vitro and in vivo models of Nf2-mutant schwannoma and in human patients (James et al, 2009; Giovannini et al, 2014; Karajannis et al 2021, others), which we have now specifically referenced in introducing these experiments. If the concern is over the distinction between our studies and that of Beauchamp et al (2021) in meningioma, we note that they showed that everolimus (20nM), which blocks pS6 in meningioma cells, also blocked Nrg1 mRNA expression, while in our hands an even higher dose of everolimus (100nM) did not reduce Nrg1 mRNA in our cells and conditions. Nevertheless, we agree that mTOR is a complex signaling nexus that receives and transmits multiple signals with various points of feedback that are likely context- and cell-type-specific. We have modified both the Results and Discussion to specifically acknowledge this and indicate the importance of in vivo follow-up.

Heterogeneity in tumors. This comment was adequately addressed.

Reviewer #2 (Remarks to the Author):

I recommend the manuscript, "Cellular mechanisms of heterogeneity in NF2-mutant schwannoma" by Christine Chiasson-MacKenzie for publication. I appreciate the clarifications made to explaining and justifying their experimental model for studying context-dependent heterogeneity in Schwann cells. Overall, the quality of the data and clarity of the manuscript is much improved and makes significant progress toward advancing understanding how context-specific signals program Schwann cells to distinct phenotypes. It still feels like the reader needs to work hard in some places to follow the rationale and conclusions of the experiments. Some simple improvements like making the abstract more mechanistic and improving some of the figure legends (e.g. Figure 2) may help.

We thank this reviewer for her/his positive and constructive feedback. We have made a concerted effort to simplify the abstract and Figure legends as suggested.

Reviewer #3 (Remarks to the Author):

We thank this reviewer for her/his positive feedback and for improving the rigor of our manuscript.

The authors have provided a number of additional experiments and text revisions that addressed my major issues. I think the manuscript have improved and is now suitable for publication.

I have only few minor points left, listed below.

- In Fig. 1 d, 1f please explain in the figure legend how you defined the cell border

We have added this explanation to the Figure 1 legend.

- Fig. 3e. Authors need to include the non-treated control coming from the same experiment and not simply referring to the control present in Fig. 1e, 1g (the absolute values vary from experiment to experiment, as also visible by comparing these two experiments).

We have added images of non-treated controls and re-quantified the data to include the control group to Figure 3E as requested.

- In Fig. 8b led is not clear to me why the Nrg+ cells are not pink. Please add more explanations in the figure legend.

We have explained, in the Figure legend, that the insets provided are images of the HALO generated segmentation filter of the Nrg+ cells, which are pseudocolored green. We have re-added the original immunofluorescence insets for Nrg1 and TGFa to Supplemental Fig. 9 to show the original staining (magenta for Nrg1; green for TGFa).

- When explaining radius there is still a "uM" led at line 632 that need to be changed to um.

Thank you for catching this! We have corrected it.